# FVBench: Benchmarking Deepfake Video Detection Capability of Large Multimodal Models

## Abstract

As generative models rapidly evolve, the realism of AI-generated videos has reached new levels, posing significant challenges for detecting the authenticity of videos. Existing deepfake detection techniques generally rely on training datasets with limited generation methods and content diversity, which limits their generalization ability on more realistic content, particularly that produced by the latest generative models. Recently, large multimodal models (LMMs) have demonstrated remarkable zero-shot performance across a variety of vision tasks. Yet, their ability to discern deepfake videos remains largely untested. To this end, we propose **FVBench**, a comprehensive deep**f**ake **v**ideo **bench**mark designed to advance video deepfake detection. It includes: **(i)** extensive content diversity, with over 120K videos covering real, AI-edited, and fully AI-generated categories, **(ii)** comprehensive model coverage, with fake videos generated and edited by 42 of the state-of-the-art video synthesis and editing models, and **(iii)** deepfake video detection benchmark for LMMs, which is a comprehensive benchmark for exploring the deepfake video detection capabilities of LMMs. The FVBench dataset and evaluation code will be publicly available upon publication, offering a valuable resource for advancing deepfake detection.

## 1 Introduction

The rapid evolution of generative models has substantially increased the realism of AI-generated videos, posing critical challenges for detecting digital content authenticity (Wang et al., 2025c; Zhang et al., 2024d; Hou et al., 2024). Traditional deepfake detection techniques typically rely on datasets with a limited number of generative models and relatively narrow content diversity (Chen et al., 2024b; Bai et al., 2024). As a result, these models are often trained to detect artifacts specific to a small set of manipulations. However, as the complexity and realism of AI-generated content continue to grow (Wang et al., 2025b;d; Chen et al., 2024d), traditional models struggle to keep up, leading to a decline in their ability to effectively detect modern deepfakes. Moreover, many of the datasets used for training detection models are often outdated and no longer reflective of the state-of-the-art generative capabilities (Khalid et al., 2021; Kuckreja et al., 2024; Yan et al., 2024), making these models less reliable in real-world scenarios. Large Multimodal Models (LMMs) have demonstrated impressive zero-shot capabilities across a wide range of vision tasks, such as face recognition, object detection, and video captioning (Yang et al., 2025; Wang et al., 2025e; Xu et al., 2025). These models have shown great potential to generalize across various tasks without the need for task-specific fine-tuning (Bai et al., 2025; Li et al., 2024a;b). However, their potential for deepfake detection remains largely unexplored.

Current deepfake video detection atasets and benchmarks for deepfake video detection suffer from several critical shortcomings that limit their practical utility and generalization: (1) **Limited content diversity**: most existing datasets concentrate primarily on facial forgeries, neglecting the growing risk of non-facial manipulations (Kuckreja et al., 2024; Felouat et al., 2024; Yan et al., 2024). Furthermore, many datasets are constructed under a binary real-or-fake paradigm (Hou et al., 2024; Khalid et al., 2021; Zhang et al., 2024a), which lack partially AI-edited content where specific regions are manipulated (Zhang et al., 2024c; Feng et al., 2024; Cohen et al., 2024). Additionally, real videos often lack the natural distortions (e.g., compression artifacts, motion blur) commonly

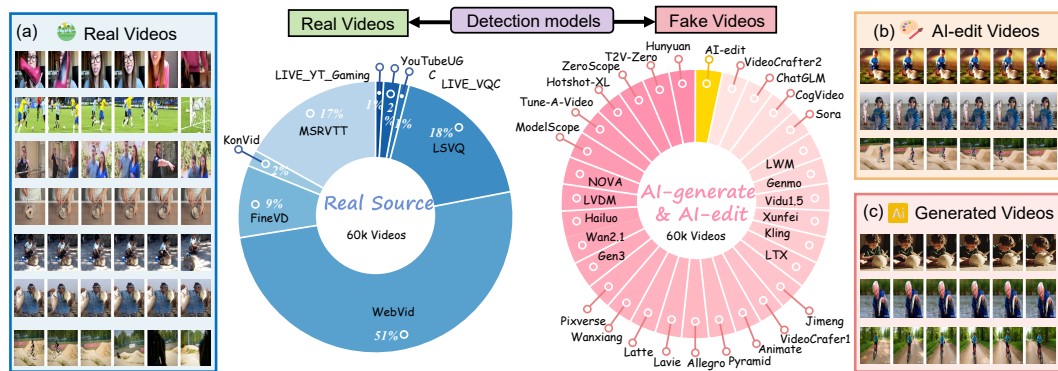

Figure 1: We present the FVBench, the large dataset for benchmarking deepfake video detection capabilities. (a) 60K real videos are collected from 8 sources. (b) 4K AI-edited videos using 12 editing models. (c) 60K fake videos are generated using 30 state-of-the-art generation models.

found in real-world scenarios (Hosu et al., 2017; Sinno & Bovik, 2018; Duan et al., 2025), which could improve robustness and generalization in detection models. (2) **Narrow model coverage**: existing datasets rely on a small number of generative models (Xie et al., 2022; Hou et al., 2024; Khalid et al., 2021), resulting in detection models learning model-specific artifacts rather than generalizable features indicative of manipulation. Additionally, older generative models often produce visible distortions, unnatural textures, or structural inconsistencies, making detection easier but hindering generalization to newer, more complex fakes generated by state-of-the-art models (Ma et al., 2025; Wang et al., 2025a; Team, 2024a;e). (3) **Restricted evaluation targets**: current benchmarks mainly assess specialized deepfake detection models, overlooking the emerging potential of LMMs in deepfake detection (Zhang et al., 2024d;a).

To this end, we introduce **FVBench**, a comprehensive deep**f**ake **v**ideo **bench**mark designed to overcome the limitations of existing datasets through three key contributions: (1) enhanced content diversity: FVBench includes not only fully synthetic, but also **partially manipulated videos** where only specific regions are edited. Furthermore, real videos incorporate natural distortions to enhance model robustness under realistic conditions. (2) expanded model coverage: fake videos include AI-generation videos using **30 models** and partial-AI videos edited by **12 AI-editing models**, covering a wide range of generation contents. (3) comprehensive evaluation framework: FVBench supports the evaluation of both the detection ability of conventional detectors and the LMMs. As illustrated in Figure 1, FVBench contains particularly deceptive examples that challenge deepfake detection models. In addition, Table 1 emphasizes the benchmark's strengths in terms of dataset size, diversity, and the comprehensiveness of its evaluation framework when compared to existing resources.

In summary, our main contributions are:

- We introduce **FVBench**, the **largest** benchmark for deepfake video detection, including generation videos from 30 models, AI-edited videos from 12 models, and 8 real sources.

- We explore **LMMs** for deepfake video detection, conducting comprehensive benchmarks that assess their performance in detecting deepfakes across various video generation methods and content types.

- Through comprehensive experiments, we find the main challenge in current detection systems is the **zero-shot generalization ability** on previously unseen generation models. While detection models can achieve high performance on known generation models, the ability to generalize to unseen models remains a significant challenge.

## 2 RELATED WORK

A variety of datasets have been developed to advance deepfake video detection. Early efforts such as UADFV (Yang et al., 2019) and FaceForensics++ (Rossler et al., 2019) focused on facial forgeries. VFHQ (Xie et al., 2022), INDIFACE (Kuckreja et al., 2024), and FakeHumanVid (Zhang et al., 2024a) target "human-centric" forgeries, covering high-quality face swapping, specific ethnic faces,

Table 1: An overview of deepfake video detection datasets.

| Dataset | Video Content | AI Generation Category | | Public Availability | Database Real Sources | AI Models | Fake Videos | Total Videos |
|---|---|---|---|---|---|---|---|---|
| | | Fully AI | Partial AI | | | | | |
| UADFV (Yang et al., 2019) | Face | ✓ | ✗ | ✓ | YouTube | 1 | 252 | 493 |
| FaceForensics++ (Rossler et al., 2019) | Face | ✗ | ✓ | ✓ | YouTube | 4 | 4000 | 5000 |
| VFHQ (Xie et al., 2022) | Face | ✗ | ✓ | ✓ | FFHQ Dataset | 1 | 8000 | 16,000 |
| INDIFACE (Kuckreja et al., 2024) | Face (Indian) | ✓ | ✗ | ✓ | YouTube | 2 | 1,668 | 2,072 |
| eKYC-DF (Felouat et al., 2024) | Face | ✓ | ✗ | ✓ | Private (Volunteers) | 3 | 12,000 | 228,000 |
| DF40 (Yan et al., 2024) | Face | ✓ | ✗ | ✓ | YouTube | Unknown | 400,000 | 800,000 |
| VIDEOSHAM (Mittal et al., 2022) | General | ✗ | ✓ | ✓ | Hollywood movies | - | 413 | 826 |
| PolyGlotFake (Hou et al., 2024) | Multimodal | ✓ | ✗ | ✓ | VoxCeleb 2 | 1 | 14,472 | 15,238 |
| FakeAVCeleb (Khalid et al., 2021) | Multimodal | ✓ | ✗ | ✓ | VoxCeleb2 | 1 | 19,500 | 20,000 |
| FakeHumanVid (Zhang et al., 2024a) | Human-centric | ✓ | ✗ | ✓ | TikTok, HDTF | 9 | 7,600 | 15,000 |
| IVY-FAKE (Zhang et al., 2024d) | General | ✓ | ✓ | ✓ | GenVideo, LOKI, YouTube | 22 | 40,000 | 73,667 |
| **FVBench (Ours)** | **General** | **✓** | **✓** | **✓** | **8 Datasets** | **42** | **62,357** | **121,902** |

and full-body generation. FakeAVCeleb (Khalid et al., 2021) and PolyGlotFake (Hou et al., 2024) extend the challenge to multimodal domains by exploring audio-visual and multilingual forgeries, while IVY-FAKE (Zhang et al., 2024d) introduces a unified benchmark for explainable detection. Yet, critical gaps persist in existing benchmarks. Many datasets rely on a small or out-of-date set of generative models, making it difficult to generalize to more advanced model generation contents. Most benchmarks also focus on completely fake videos, ignoring the common issue of partially AI-edited content. Their collections of real videos are often pristine, lacking the natural distortions of real-world content and thus limiting the robustness of detection models. FVBench stands out for its scale, diversity, and balanced inclusion of real, AI-edited, and fully AI-generated videos from state-of-the-art models.

## 3 DATABASE CONSTRUCTION

### 3.1 REAL VIDEO COLLECTION

To ensure content diversity and realism, FVBench incorporates real videos from eight well-known public natural video datasets. These datasets are widely recognized for their diverse content and high-quality annotations, providing a solid foundation for deepfake detection across a broad range of scenarios. The MSRVTT (Xu et al., 2016) dataset includes 10,000 videos spanning various activities and is typically used for video-to-text tasks, while KonVid (Hosu et al., 2017) focuses on video quality assessment with 1,200 clips that capture various video distortions. FineVD (Duan et al., 2025) provides 5,074 videos with fine-grained annotations of distortions like noise and compression artifacts, ideal for training models on video quality degradation. The WebVid (Bain et al., 2021) dataset is scraped from the web, covering diverse content types like user-generated videos and news, making it perfect for video retrieval and action recognition tasks. LSVQ (Ying et al., 2021), with 10,759 video clips, offers real-world content for perceptual quality assessment, while LIVEVQC (Sinno & Bovik, 2018) focuses on videos impacted by network distortions in streaming scenarios. Additionally, YouTubeUGC (Wang et al., 2019) contributes 1,147 user-generated videos from the YouTube platform, covering a wide range of genres and providing rich content for scene detection and video quality tasks. Lastly, the LIVE-YT-Gaming (Yu et al., 2023) dataset, consisting of 600 gaming videos, caters to the gaming content genre. Collectively, these datasets ensure that FVBench includes a diverse mix of real-world videos, capturing a broad spectrum of quality, distortions, and content types to challenge deepfake detection models.

### 3.2 AI-EDITING VIDEO COLLECTION

We collect 180 base videos from Kinetics-400 (Kay et al., 2017) and DAVIS (Pont-Tuset et al., 2018) (50% human actions, 15% animal behaviors, 35% other). Editing prompts are generated using DeepSeek-R1 (DeepSeek-AI, 2025), covering five key tasks: color, action, background, object operation, and style change (e.g., oil painting, ink-style). These prompts were engineered to maintain 60% of the original content's semantics, ensuring focused edits. We then use 12 open-source, diffusion-based video editing models including Tune-A-Video (Wu et al., 2023), TokenFlow (Qu et al., 2023), CCEdit (Feng et al., 2024), ControlVideo (Zhang et al., 2024c), FateZero (Qi et al., 2023), FLATTEN (Cong et al., 2024), FRESCO (Yang et al., 2024a), Pix2Video (Ceylan et al., 2023), RAVE (Kara et al., 2024), SlicEdit (Cohen et al., 2024), and Vid2Vid-Zero (Wang et al., 2024). Finally, we obtained 3,857 valid AI-edited videos.

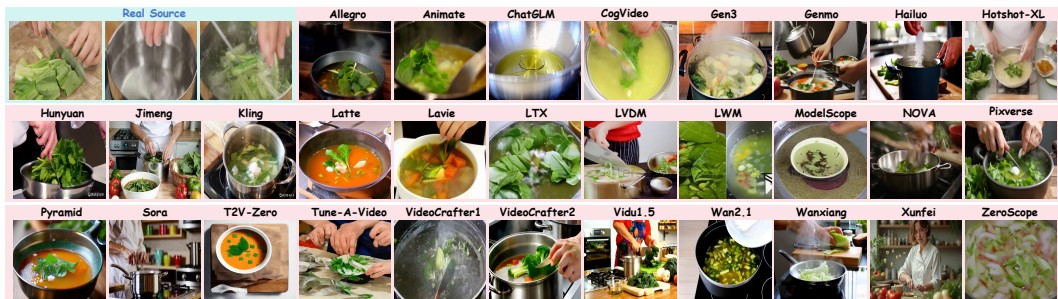

Figure 2: Visualization of video frames in the FVBench dataset.

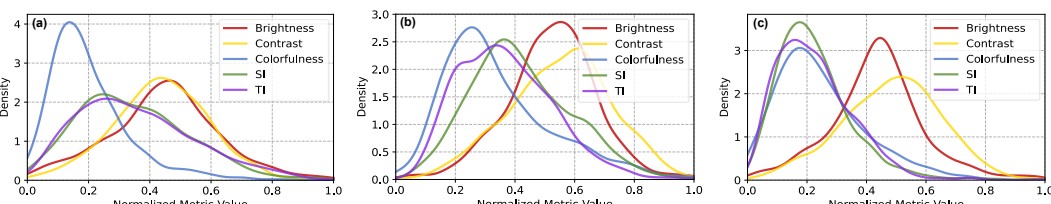

Figure 3: Feature distribution of the FVBench. (a) Feature distribution of real videos. (b) Feature distribution of AI-edited videos. (c) Feature distribution of AI-generated videos.

### 3.3 FAKE VIDEO GENERATION

To construct a diverse and challenging set of fake videos, we utilize 30 state-of-the-art video generation models, including 18 open-source generation models: Pyramid (Jin et al., 2024), Wan2.1 (Wang et al., 2025a), Allegro (Zhou et al., 2024), VideoCrafter2 (Chen et al., 2024a), CogVideo X1.5 (Yang et al., 2024b), Animate (Xu et al., 2024), Lavie (Wang et al., 2023b), Hotshot-XL (Mullan et al., 2023), Latte (Ma et al., 2025), VideoCrafter1 (Chen et al., 2023), Text2Video-Zero (Khachatryan et al., 2023), ModelScope (Wang et al., 2023a), Tune-A-Video (Wu et al., 2023), LTX (HaCohen et al., 2024), LVDM (He et al., 2022), ZeroScope (Team, 2024h), and LWM (Liu et al., 2024a) and 12 close-source generation models: Pixverse (AI, 2024), Wanxiang (Cloud, 2024), Hailuo (Team, 2024d), Jimeng (Team, 2024a), Hunyuan (Li et al., 2024c), Sora (Team, 2024e), Vidu1.5 (Team, 2024f), Gen3 (Runway, 2024), Kling (Team, 2024c), Genmo (Team, 2024b), ChatGLM (GLM et al., 2024), and Xunfei (Team, 2024g). To guarantee fairness, all generative models are used with their official default weights, with no additional adaptation or tuning.

The video prompts are mostly obtained from 8 existing open-domain text-video pair datasets, with some being refined using DeepSeek R1 (Guo et al., 2025) to ensure clarity and diversity. We use 2,750 distinct prompts in the training set; each prompt is processed by 18 open-source models. The test set includes 300 distinct prompts created by all 30 models. This approach generates 58,500 videos from 3,050 distinct prompts (2,750 prompts x 18 open-source models + 300 prompts x 30 models). The imbalance between the training and testing sets is due to two factors: (1) producing videos using close-source tools is expensive, and (2) we want to test the scalability of evaluation metrics on training-set unseen generation models. Figure 7 shows that all 30 models are given the same set of prompts based on real-world video captions. Some close-source models, such as Kling (Team, 2024c) and Hailuo (Team, 2024d), can provide highly detailed outputs that even surpass the real source.

### 3.4 DATABASE ANALYSIS

As shown in Figure 3, we analyze the feature distribution of real, AI-edited, and AI-generated videos in the FVBench dataset across five video quality-related features: colorfulness, brightness, contrast, spatial information (SI), and temporal information (TI). The analysis reveals that AI-generated videos display the highest values for SI and TI, indicating their rich spatial and temporal detail. In contrast, real videos exhibit greater colorfulness. AI-edited videos, which combine both authentic and manipulated content, show feature values that lie between the real and fake videos.

Table 2: **Performance benchmark on AI-generated video subsets.** ♡Conventional deepfake detection models, ★open-source and △close-source LMMs. ♦* refers to finetuned models. We **bold** the best results.

| Methods / Datasets | Genmo | Hailuo | T2V-Zero | Tune-A-Video | LVDM | LWM | LTX | ZeroScope | Jimeng | VCrafter2 |
|---|---|---|---|---|---|---|---|---|---|---|
| ♡Swin3D_T | 74.00% | 76.33% | 67.67% | 56.67% | 66.00% | 53.67% | 61.00% | 36.67% | 68.67% | 68.67% |
| ♡ResNet3D_18 | 82.00% | 93.00% | 95.00% | 72.67% | 73.00% | **98.33%** | 78.67% | 77.67% | 92.33% | 85.67% |
| ♡AIGVDet | 83.74% | 66.82% | 91.49% | 88.65% | 62.78% | 71.85% | 59.32% | 57.56% | 48.56% | 54.28% |
| ♡DeMamba | 4.33% | 0.33% | 0.00% | 2.00% | 0.00% | 10.67% | 6.33% | 2.67% | 0.00% | 0.33% |
| ★Llava-one-vision (0.5B) | 49.00% | 41.00% | 48.33% | 47.67% | 51.00% | 48.67% | 50.00% | 48.00% | 40.67% | 49.00% |
| ★InternVL2.5 (1B) | 54.33% | 50.33% | 57.33% | 58.00% | 59.67% | 59.00% | 48.33% | 49.00% | 47.33% | 51.67% |
| ★InternVL3 (1B) | 70.00% | 56.00% | 70.00% | 65.67% | 65.33% | 67.33% | 64.67% | 63.33% | 60.00% | 62.33% |
| ★Qwen2.5-VL (3B) | 95.00% | **98.33%** | 82.67% | 87.00% | 79.33% | 80.67% | 78.33% | 79.00% | 93.67% | 87.00% |
| ★VideoLlava (7B) | 50.67% | 49.00% | 49.67% | 50.00% | 48.00% | 46.33% | 51.00% | 50.33% | 45.67% | 49.00% |
| ★Llava-one-vision (7B) | 75.33% | 38.00% | 42.33% | 60.33% | 76.67% | 63.67% | 51.67% | 48.00% | 25.33% | 36.33% |
| ★mPLUG-Owl3 (7B) | 86.67% | 77.00% | 74.00% | 76.33% | 94.33% | 94.00% | 83.33% | 84.33% | 51.00% | 80.00% |
| ★Qwen2.5-VL (7B) | 89.33% | 86.67% | 82.00% | 77.00% | 64.67% | 64.00% | 62.33% | 65.33% | 77.67% | 60.67% |
| ★InternLM-XComposer2.5 (7B) | 95.67% | 95.67% | 90.33% | 94.00% | 95.00% | 89.67% | **94.67%** | **97.67%** | 88.33% | **94.67%** |
| ★VideoLlama3 (8B) | 85.00% | 78.33% | 71.00% | 48.33% | 67.33% | 61.67% | 77.00% | 68.33% | 78.00% | 59.00% |
| ★LLaVA-NeXT-Video (8B) | 53.33% | 54.33% | 53.67% | 54.67% | 66.00% | 70.67% | 62.67% | 67.00% | 43.00% | 51.00% |
| ★InternVL2.5 (8B) | 88.00% | 90.33% | 75.00% | 80.00% | 69.67% | 71.67% | 72.00% | 75.67% | 79.33% | 76.00% |
| ★InternVL3 (9B) | 89.00% | 83.67% | 77.00% | 81.00% | 78.33% | 74.33% | 80.00% | 73.67% | 81.67% | 76.00% |
| ★Llama3.2-Vision (11B | 84.00% | 79.00% | 79.00% | 71.67% | 76.33% | 66.33% | 77.33% | 73.67% | 64.00% | 82.33% |
| ★InternVL2.5 (26B) | 60.67% | 72.67% | 61.33% | 63.67% | 69.33% | 64.33% | 63.33% | 64.00% | 60.33% | 60.33% |
| ★Qwen2.5-VL (32B) | 76.67% | 80.33% | 71.67% | 69.00% | 45.33% | 56.00% | 54.00% | 65.67% | 70.67% | 65.33% |
| ★InternVL2.5 (38B) | 70.67% | 67.33% | 67.67% | 69.67% | 62.00% | 69.67% | 63.00% | 64.67% | 60.33% | 62.00% |
| ★InternVL3 (38B) | 81.00% | 84.33% | 80.67% | 79.33% | 74.00% | 74.33% | 70.67% | 67.67% | 81.67% | 64.33% |
| ★Qwen2.5-VL (72B) | 93.00% | 93.00% | 80.33% | 71.33% | 51.33% | 54.33% | 51.33% | 52.33% | 79.33% | 57.00% |
| ★InternVL3 (78B) | 69.00% | 80.00% | 72.00% | 76.00% | 66.67% | 72.67% | 63.67% | 54.67% | 68.00% | 57.33% |
| △Gemini1.5-pro | 98.33% | 81.67% | **98.33%** | **98.33%** | **95.00%** | 81.67% | 75.00% | 66.67% | **95.00%** | 91.67% |
| △GPT4o | **100.0%** | 65.00% | 86.67% | 88.33% | 71.67% | 76.67% | 68.33% | 46.67% | 46.67% | 93.33% |
| **Model Average (Zero-shot)** | 73.35% | 70.49% | 68.34% | 66.69% | 65.09% | 65.99% | 63.53% | 61.95% | 63.15% | 62.09% |
| ♦Swin3D_T* | 100.0% | 100.0% | 100.0% | 100.0% | 100.0% | 100.0% | 100.0% | 100.0% | 100.0% | 100.0% |
| ♦ResNet3D_18* | 100.0% | 100.0% | 100.0% | 100.0% | 100.0% | 100.0% | 100.0% | 100.0% | 100.0% | 100.0% |
| ♦InternVL2.5 (8B)* | 100.0% | 100.0% | 100.0% | 100.0% | 100.0% | 100.0% | 100.0% | 100.0% | 100.0% | 100.0% |
| ♦InternVL3 (9B)* | 100.0% | 100.0% | 100.0% | 100.0% | 100.0% | 100.0% | 100.0% | 100.0% | 100.0% | 100.0% |

| Methods / Datasets | Xunfei | Hotshot | Hunyuan | CogVideo | NOVA | VCrafter1 | Sora | ChatGLM | Wan2.1 | Animate |
|---|---|---|---|---|---|---|---|---|---|---|
| ♡Swin3D_T | 83.67% | 52.33% | 73.00% | 60.00% | 63.67% | 58.00% | 64.67% | 61.00% | 61.33% | 66.33% |
| ♡ResNet3D_18 | 86.00% | 71.00% | 74.00% | 84.67% | 78.67% | **97.33%** | 85.33% | 89.67% | 89.67% | 86.67% |
| ♡AIGVDet | 49.45% | 53.24% | 71.51% | 21.57% | 72.48% | 21.30% | 76.45% | 46.67% | 74.35% | 23.81% |
| ♡DeMamba | 0.67% | 8.33% | 0.33% | 13.67% | 13.67% | 0.00% | 0.67% | 0.00% | 1.00% | 16.67% |
| ★Llava-one-vision (0.5B) | 48.33% | 48.00% | 42.33% | 47.33% | 47.67% | 48.33% | 45.67% | 46.33% | 45.67% | 46.67% |
| ★InternVL2.5 (1B) | 52.67% | 49.33% | 42.33% | 51.67% | 45.33% | 56.33% | 49.33% | 48.00% | 48.00% | 48.67% |
| ★InternVL3 (1B) | 58.33% | 62.67% | 51.67% | 58.00% | 55.67% | 57.00% | 59.00% | 59.67% | 57.33% | 56.67% |
| ★Qwen2.5-VL (3B) | 83.33% | 71.00% | 82.67% | 72.67% | 68.33% | 64.00% | 70.33% | 66.67% | 75.67% | 47.33% |
| ★VideoLlava (7B) | 45.67% | 47.33% | 46.00% | 50.00% | 48.67% | 48.00% | 48.67% | 48.33% | 46.33% | 48.33% |
| ★Llava-one-vision (7B) | 23.33% | 32.00% | 16.00% | 30.67% | 36.33% | 35.67% | 20.00% | 30.33% | 19.67% | 23.00% |
| ★mPLUG-Owl3 (7B) | 59.33% | 68.67% | 51.67% | 62.67% | 71.67% | 72.33% | 45.67% | 52.67% | 42.33% | 54.33% |
| ★Qwen2.5-VL (7B) | 52.33% | 41.00% | 48.67% | 45.67% | 38.33% | 40.33% | 50.00% | 46.67% | 43.67% | 32.33% |
| ★InternLM-XComposer2.5 (7B) | 90.33% | 99.00% | 84.00% | 95.33% | 96.33% | 95.67% | 93.67% | 90.67% | 92.00% | 93.00% |
| ★VideoLlama3 (8B) | 64.33% | 60.67% | 78.00% | 67.00% | 63.67% | 55.33% | 68.00% | 65.67% | 63.33% | 58.33% |
| ★LLaVA-NeXT-Video (8B) | 46.67% | 51.67% | 44.67% | 51.67% | 51.33% | 56.33% | 47.67% | 46.33% | 43.00% | 44.33% |
| ★InternVL2.5 (8B) | 67.67% | 63.67% | 78.33% | 63.33% | 63.67% | 53.33% | 61.33% | 62.33% | 61.33% | 62.33% |
| ★InternVL3 (9B) | 68.33% | 69.00% | 70.00% | 71.67% | 70.00% | 67.00% | 64.67% | 66.67% | 68.67% | 58.00% |
| ★Llama3.2-Vision (11B) | 81.00% | 77.00% | 72.67% | 74.67% | 73.33% | 79.33% | 77.67% | 83.67% | 79.67% | 67.33% |
| ★InternVL2.5 (26B) | 50.33% | 56.67% | 59.33% | 56.33% | 58.67% | 63.00% | 47.33% | 49.33% | 48.33% | 57.00% |
| ★Qwen2.5-VL (32B) | 64.33% | 46.33% | 48.67% | 39.67% | 37.00% | 47.33% | 52.33% | 39.00% | 44.67% | 31.67% |
| ★InternVL2.5 (38B) | 48.00% | 51.33% | 53.00% | 53.00% | 50.00% | 41.33% | 39.67% | 47.00% | 47.67% | 44.00% |
| ★InternVL3 (38B) | 66.67% | 64.33% | 64.67% | 63.67% | 65.33% | 53.67% | 59.33% | 62.00% | 64.67% | 52.00% |
| ★Qwen2.5-VL (72B) | 41.00% | 41.67% | 50.33% | 34.00% | 33.67% | 35.00% | 59.33% | 35.67% | 43.67% | 27.00% |
| ★InternVL3 (78B) | 47.00% | 49.00% | 55.00% | 50.67% | 44.67% | 49.33% | 42.00% | 49.00% | 44.00% | 41.00% |
| △Gemini1.5-pro | 58.33% | 68.33% | 83.33% | 43.33% | 66.67% | 73.33% | 85.00% | 35.00% | 46.67% | 25.00% |
| △GPT4o | 45.00% | 30.00% | 26.67% | 30.00% | 46.67% | 36.67% | 16.67% | 26.67% | 35.00% | 15.00% |
| **Model Average (Zero-shot)** | 57.45% | 55.64% | 56.62% | 54.98% | 56.17% | 53.97% | 55.37% | 50.39% | 54.42% | 49.45% |
| ♦Swin3D_T* | 100.0% | 100.0% | 100.0% | 100.0% | 100.0% | 100.0% | 100.0% | 100.0% | 100.0% | 100.0% |
| ♦ResNet3D_18* | 100.0% | 100.0% | 100.0% | 100.0% | 100.0% | 100.0% | 100.0% | 100.0% | 100.0% | 100.0% |
| ♦InternVL2.5 (8B)* | 100.0% | 100.0% | 100.0% | 100.0% | 100.0% | 100.0% | 100.0% | 100.0% | 100.0% | 100.0% |
| ♦InternVL3 (9B)* | 100.0% | 100.0% | 100.0% | 100.0% | 100.0% | 100.0% | 100.0% | 100.0% | 100.0% | 100.0% |

| Methods / Datasets | Wanxiang | Allegro | Pyramid | Vidu1.5 | Lavie | Kling | Pixverse | Latte | MScope | Gen3 | Overall |
|---|---|---|---|---|---|---|---|---|---|---|---|
| ♡Swin3D_T | 75.33% | 73.00% | 78.33% | 75.33% | 58.33% | 66.33% | 68.67% | 58.33% | 53.67% | 70.52% | 65.04% |
| ♡ResNet3D_18 | 80.67% | 85.33% | 83.00% | 79.33% | 82.00% | 81.33% | 85.00% | 79.67% | 72.00% | 85.01% | 80.85% |
| ♡AIGVDet | 39.23% | 91.07% | 0.00% | 42.45% | 48.97% | 46.32% | 53.45% | 68.95% | 59.56% | 67.59% | 57.12% |
| ♡DeMamba | 0.33% | 3.00% | 2.33% | 6.00% | 3.00% | 1.33% | 0.33% | 6.00% | 0.00% | 0.40% | 3.30% |
| ★Llava-one-vision (0.5B)) | 48.00% | 48.00% | 47.33% | 49.00% | 46.00% | 47.33% | 47.00% | 48.00% | 50.33% | 47.41% | 47.27% |
| ★InternVL2.5 (1B) | 44.67% | 41.67% | 46.67% | 47.33% | 53.67% | 43.33% | 49.00% | 52.33% | 57.00% | 50.00% | 50.41% |
| ★InternVL3 (1B) | 57.67% | 63.00% | 58.00% | 53.00% | 67.00% | 53.00% | 55.67% | 59.00% | 67.33% | 59.36% | 60.66% |
| ★Qwen2.5-VL (3B) | **97.33%** | 87.67% | 85.00% | 87.00% | 72.00% | 92.00% | 76.67% | 71.33% | 71.33% | 86.65% | 79.67% |
| ★VideoLlava (7B) | 47.67% | 50.33% | 46.33% | 46.00% | 52.67% | 49.00% | 42.00% | 47.67% | 48.33% | 49.80% | 48.23% |
| ★Llava-one-vision (7B) | 23.00% | 33.33% | 32.00% | 36.67% | 38.67% | 22.00% | 22.00% | 27.33% | 36.67% | 20.12% | 35.88% |
| ★mPLUG-Owl3 (7B) | 66.00% | 75.33% | 61.00% | 77.67% | 61.00% | 61.67% | 61.67% | 64.33% | 75.67% | 49.20% | 68.03% |
| ★Qwen2.5-VL (7B) | 55.33% | 64.00% | 67.00% | 62.00% | 53.33% | 52.67% | 51.67% | 49.67% | 38.67% | 54.58% | 57.10% |
| ★InternLM-XComposer2.5 (7B) | 89.67% | **93.00%** | **93.67%** | 93.33% | 93.00% | 90.00% | **91.67%** | **91.00%** | **94.00%** | 94.42% | 92.98% |
| ★VideoLlama3 (8B) | 75.00% | 69.67% | 66.67% | 76.00% | 65.00% | 75.33% | 65.00% | 66.33% | 58.00% | 60.16% | 67.01% |
| ★LLaVA-NeXT-Video (8B) | 47.00% | 53.33% | 46.67% | 49.00% | 53.00% | 44.67% | 46.33% | 49.00% | 59.67% | 49.80% | 51.95% |
| ★InternVL2.5 (8B) | 85.67% | 73.33% | 65.33% | 76.33% | 68.00% | 81.33% | 67.00% | 67.00% | 61.67% | 65.34% | 70.97% |
| ★InternVL3 (9B) | 81.00% | 70.00% | 73.00% | 79.67% | 72.00% | 78.33% | 72.33% | 74.00% | 74.00% | 70.72% | 73.79% |
| ★Llama3.2-Vision (11B) | 77.00% | 78.33% | 77.33% | 79.33% | 78.33% | 78.67% | 82.33% | 79.00% | 77.67% | 74.30% | 77.09% |
| ★InternVL2.5 (26B) | 57.33% | 56.00% | 59.33% | 59.00% | 59.33% | 57.00% | 50.33% | 57.67% | 54.00% | 52.39% | 58.28% |
| ★Qwen2.5-VL (32B) | 59.00% | 58.33% | 68.67% | 50.00% | 57.33% | 41.00% | 64.67% | 48.67% | 44.67% | 59.36% | 55.25% |
| ★InternVL2.5 (38B) | 64.00% | 54.00% | 51.33% | 54.67% | 54.00% | 52.00% | 52.33% | 54.33% | 58.00% | 44.82% | 55.72% |
| ★InternVL3 (38B) | 70.67% | 73.33% | 75.33% | 62.00% | 62.33% | 65.33% | 64.00% | 62.33% | 64.00% | 65.74% | 67.98% |
| ★Qwen2.5-VL (72B) | 62.33% | 55.00% | 65.00% | 50.00% | 50.00% | 58.33% | 58.67% | 50.33% | 32.33% | 57.57% | 54.14% |
| ★InternVL3 (78B) | 57.00% | 57.67% | 62.33% | 59.00% | 60.00% | 53.67% | 51.67% | 54.33% | 49.33% | 53.78% | 57.02% |
| △Gemini1.5-pro | 90.00% | 58.33% | 76.67% | 85.00% | 71.67% | **98.33%** | 58.33% | 58.33% | 70.00% | 38.33% | 71.15% |
| △GPT4o | 36.67% | 43.33% | 70.00% | 38.33% | 53.33% | 50.00% | 50.00% | 53.67% | 46.67% | 28.33% | 49.86% |
| **Model Average (Zero-shot)** | 60.87% | 62.82% | 58.82% | 60.23% | 58.71% | 58.00% | 57.81% | 57.78% | 56.58% | 57.88% | 59.82% |
| ♦Swin3D_T* | 100.0% | 100.0% | 100.0% | 100.0% | 100.0% | 100.0% | 100.0% | 100.0% | 100.0% | 100.0% | 100.0% |
| ♦ResNet3D_18* | 100.0% | 100.0% | 100.0% | 100.0% | 100.0% | 100.0% | 100.0% | 100.0% | 100.0% | 100.0% | 100.0% |
| ♦InternVL2.5 (8B)* | 100.0% | 100.0% | 100.0% | 100.0% | 100.0% | 100.0% | 100.0% | 100.0% | 100.0% | 100.0% | 100.0% |
| ♦InternVL3 (9B)* | 100.0% | 100.0% | 100.0% | 100.0% | 100.0% | 100.0% | 100.0% | 100.0% | 100.0% | 100.0% | 100.0% |

Table 3: **Performance benchmark on real video subsets.** ♡Conventional deepfake detection models, ★open-source and △close-source LMMs. ♦* refers to finetuned models. We **bold** the best results.

| Methods / Datasets | MSRVTT | KoNViD | FineVD | WebVid | LSVQ | LIVEVQC | YouTubeUGC | LIVE-YT-Gaming | Overall |
|---|---|---|---|---|---|---|---|---|---|
| ♡Swin3D_T | 58.95% | 54.58% | 67.83% | 70.97% | 55.86% | 63.25% | 73.25% | 80.00% | 65.47% |
| ♡ResNet3D_18 | 84.45% | 85.83% | 86.78% | 94.12% | 82.85% | 90.60% | 89.04% | 82.50% | 89.56% |
| ♡AIGVDet | **100.0%** | 98.75% | 93.97% | 89.86% | 93.12% | 93.16% | 84.84% | 89.17% | 92.74% |
| ♡DeMamba | 99.90% | 98.75% | **99.41%** | **99.95%** | 84.34% | **100.0%** | **100.0%** | **98.33%** | **97.03%** |
| ★Llava-one-vision (0.5B)) | 59.35% | 62.50% | 55.91% | 50.25% | 58.32% | 60.68% | 57.89% | 50.00% | 56.86% |
| ★InternVL2.5 (1B) | 68.80% | 59.17% | 49.90% | 67.60% | 58.27% | 54.70% | 59.21% | 54.17% | 58.98% |
| ★InternVL3 (1B) | 47.80% | 45.42% | 46.46% | 51.41% | 46.33% | 51.28% | 43.42% | 52.50% | 48.08% |
| ★Qwen2.5-VL (3B) | 53.00% | 79.58% | 27.85% | 58.66% | 72.15% | 82.91% | 52.19% | 45.00% | 58.92% |
| ★VideoLlava (7B) | 59.30% | 57.92% | 48.72% | 56.67% | 60.69% | 54.82% | 54.45% | 35.00% | 53.45% |
| ★Llava-one-vision (7B) | 80.25% | 96.25% | 62.40% | 86.67% | 91.03% | 100.00% | 69.74% | 45.00% | 78.92% |
| ★mPLUG-Owl3 (7B) | 47.35% | 78.33% | 19.98% | 78.32% | 78.39% | 91.45% | 48.68% | 2.50% | 55.63% |
| ★Qwen2.5-VL (7B) | 70.80% | 93.75% | 49.21% | 62.78% | 88.29% | 63.16% | 83.43% | 15.83% | 65.91% |
| ★InternLM-XComposer2.5 (7B) | 65.75% | 76.67% | 51.87% | 62.73% | 72.58% | 65.81% | 45.61% | 55.83% | 62.11% |
| ★VideoLlama3 (8B) | 50.45% | 59.17% | 29.63% | 41.67% | 57.48% | 39.04% | 50.23% | 30.00% | 44.71% |
| ★LLaVA-NeXT-Video (8B) | 53.65% | 65.00% | 44.00% | 42.67% | 59.34% | 61.54% | 50.00% | 39.17% | 51.92% |
| ★InternVL2.5 (8B) | 52.55% | 71.67% | 38.48% | 57.93% | 67.01% | 78.63% | 49.56% | 34.17% | 56.25% |
| ★InternVL3 (9B) | 44.20% | 63.75% | 29.63% | 51.31% | 56.51% | 69.23% | 41.67% | 22.50% | 47.35% |
| ★Llama3.2-Vision (11B) | 24.05% | 38.33% | 20.47% | 25.79% | 33.04% | 50.43% | 25.44% | 14.17% | 28.97% |
| ★InternVL2.5 (26B) | 65.40% | 62.92% | 46.75% | 51.15% | 67.47% | 73.50% | 60.96% | 38.33% | 58.31% |
| ★Qwen2.5-VL (32B) | 70.05% | 96.25% | 56.89% | 85.02% | 89.50% | 99.15% | 72.90% | 30.83% | 75.07% |
| ★InternVL2.5 (38B) | 68.50% | 80.42% | 47.15% | 74.06% | 80.81% | 90.60% | 61.84% | 24.17% | 65.94% |
| ★InternVL3 (38B) | 68.60% | 84.58% | 39.96% | 77.98% | 79.04% | 91.45% | 60.53% | 12.50% | 64.33% |
| ★Qwen2.5-VL (72B) | 83.10% | 97.08% | 68.90% | 86.62% | 92.57% | 100.00% | 77.34% | 39.17% | 80.60% |
| ★InternVL3 (78B) | 75.70% | 93.33% | 57.09% | 75.71% | 87.50% | 97.44% | 72.81% | 33.33% | 74.11% |
| △Gemini1.5-pro | 90.00% | **100.0%** | 80.00% | 76.67% | 93.10% | **100.0%** | 69.57% | 50.88% | 79.91% |
| △GPT-4o | 90.00% | 96.67% | 75.00% | 88.14% | **93.22%** | **100.0%** | 78.26% | 63.33% | 72.10% |
| Model Average (Zero-shot) | 64.66% | 75.00% | 51.62% | 66.72% | 71.36% | 75.95% | 61.86% | 42.67% | 63.84% |
| ♦Swin3D_T* | 94.25% | 65.62% | 82.27% | 96.93% | 75.42% | 67.74% | 86.96% | 92.71% | 86.03% |
| ♦ResNet3D_18* | 99.88% | 100.0% | 98.65% | 100.0% | 98.78% | 100.0% | 100.0% | 100.0% | 99.41% |
| ♦InternVL2.5 (8B)* | 100.0% | 98.44% | 99.38% | 100.0% | 99.30% | 100.0% | 97.83% | 100.0% | 99.62% |
| ♦InternVL3 (9B)* | 99.69% | 97.40% | 97.41% | 86.63% | 98.19% | 99.98% | 86.96% | 100.0% | 95.23% |

# 4 BENCHMARK AND EVALUATION

We benchmark and evaluate both the in-domain performance and cross-generator generalization of various deepfake detection models across three subsets: real, AI-edited, and AI-generated videos.

## 4.1 EXPERIMENT SETUP

We evaluate the models' ability to classify real and fake videos using two standard metrics: accuracy (Acc) and F1-score. Accuracy is calculated as the proportion of correctly classified real or fake videos out of all relevant samples in the dataset.

To provide a more balanced evaluation that accounts for both precision and recall, we also compute the F1-score, which is the harmonic mean of precision and recall:

$$F1 = \frac{2 \times \text{Precision} \times \text{Recall}}{\text{Precision} + \text{Recall}} \qquad (1)$$

where precision and recall are:

$$\text{Precision} = \frac{\text{TP}}{\text{TP} + \text{FP}}, \quad \text{Recall} = \frac{\text{TP}}{\text{TP} + \text{FN}} \qquad (2)$$

where TP (True Positives) represents the number of real or fake videos correctly identified by the model, and FN (False Negatives) indicates the number of videos incorrectly classified as the opposite category. We directly use publicly available pre-trained weights to conduct inference on the test datasets. For large multimodal models (LMMs), we perform inference using a prompt-based question-answering approach. To minimize any bias in the responses, we alternate between the following two instructions: (1) "Is this a real video or a generated video? Just answer with A or B. A: real or B: generated." and (2) "Is this a generated video or a real video? Just answer with A or B. A: generated or B: real." Additionally, we fine-tune two of the LMMs with LoRA (Hu et al., 2022) (r=16), using the same 4:1 training and testing split. The fine-tuning process is conducted over 5 epochs. The models are implemented in PyTorch and trained on a 40GB NVIDIA RTX A6000 GPU with a batch size of 4. The initial learning rate is set to 1e-5 and is adjusted using a cosine annealing strategy.

## 4.2 IN-DOMAIN PERFORMANCE ON FVBENCH

We benchmark model performance on AI-generated video subsets, as shown in Table 2. We can observe that traditional deep learning-based detection models, such as DeMamba (Chen et al., 2024b)

Table 4: **Performance benchmark on AI-edit video subsets, including five editing types.** ♦* refers to finetuned models.

| Dimension | Background | | Style Change | | Color Change | | Action Edit | | Object Operation | | Overall | |
|---|---|---|---|---|---|---|---|---|---|---|---|---|
| Methods / Metrics | Acc(%)↑ | F1↑ | Acc(%)↑ | F1↑ | Acc(%)↑ | F1↑ | Acc(%)↑ | F1↑ | Acc(%)↑ | F1↑ | Acc(%)↑ | F1↑ |
| ♡Swin3D_T | 59.42 | 0.745 | 53.03 | 0.693 | 52.10 | 0.685 | 61.76 | 0.764 | 45.76 | 0.628 | 53.18 | 0.694 |
| ♡ResNet3D_18 | 86.78 | 0.929 | 85.71 | 0.923 | 87.80 | 0.935 | 85.12 | 0.92 | 86.46 | 0.927 | 86.53 | 0.928 |
| ♡AIGVDet | 82.95 | 0.820 | 84.39 | 0.827 | 73.36 | 0.771 | 82.02 | 0.816 | 76.44 | 0.787 | 79.83 | 0.804 |
| ♡DeMamba | 0.000 | 0.000 | 0.000 | 0.000 | 0.000 | 0.000 | 0.000 | 0.000 | 0.000 | 0.000 | 0.000 | 0.000 |
| ★Llava-one-vision (0.5B)) | 51.24 | 0.678 | 48.12 | 0.650 | 49.76 | 0.664 | 42.98 | 0.601 | 46.88 | 0.638 | 47.93 | 0.611 |
| ★InternVL2.5 (1B) | 58.68 | 0.740 | 58.65 | 0.739 | 52.20 | 0.686 | 52.07 | 0.685 | 52.08 | 0.685 | 54.27 | 0.659 |
| ★InternVL3 (1B) | 57.02 | 0.726 | 68.42 | 0.813 | 67.80 | 0.808 | 60.33 | 0.753 | 71.88 | 0.836 | 66.06 | 0.740 |
| ★Qwen2.5-VL (3B) | 84.30 | 0.915 | 85.71 | 0.923 | 87.80 | 0.935 | 68.60 | 0.814 | 82.29 | 0.903 | 82.51 | 0.847 |
| ★VideoLlava (7B) | 48.76 | 0.656 | 42.86 | 0.600 | 52.68 | 0.690 | 38.02 | 0.551 | 46.35 | 0.633 | 46.50 | 0.594 |
| ★Llava-one-vision (7B) | 57.85 | 0.733 | 76.69 | 0.868 | 61.46 | 0.761 | 38.02 | 0.551 | 60.94 | 0.757 | 59.72 | 0.727 |
| ★mPLUG-Owl3 (7B) | 73.55 | 0.848 | 81.95 | 0.901 | 83.90 | 0.912 | 58.68 | 0.740 | 77.60 | 0.874 | 76.42 | 0.829 |
| ★Qwen2.5-VL (7B) | 71.07 | 0.831 | 76.69 | 0.868 | 81.95 | 0.901 | 42.98 | 0.601 | 72.92 | 0.843 | 70.98 | 0.790 |
| ★InternLM-XComposer2.5 (7B) | **94.21** | **0.970** | **96.24** | **0.981** | **94.15** | **0.970** | **96.69** | **0.983** | **96.88** | **0.984** | **95.60** | **0.958** |
| ★VideoLlama3 (8B) | 70.25 | 0.825 | 82.71 | 0.905 | 82.44 | 0.904 | 59.50 | 0.746 | 76.56 | 0.867 | 75.52 | 0.799 |
| ★LLaVA-NeXT-Video (8B) | 57.02 | 0.726 | 55.64 | 0.715 | 60.49 | 0.754 | 48.76 | 0.656 | 55.73 | 0.716 | 56.09 | 0.677 |
| ★InternVL2.5 (8B) | 65.29 | 0.790 | 81.95 | 0.901 | 82.93 | 0.907 | 57.02 | 0.726 | 73.96 | 0.850 | 73.70 | 0.789 |
| ★InternVL3 (9B) | 77.69 | 0.874 | 84.96 | 0.919 | 80.98 | 0.895 | 61.16 | 0.759 | 77.08 | 0.871 | 77.07 | 0.812 |
| ★Llama3.2-Vision (11B) | 80.17 | 0.890 | 86.47 | 0.927 | 84.88 | 0.918 | 77.69 | 0.874 | 78.12 | 0.877 | 81.61 | 0.826 |
| ★InternVL2.5 (26B) | 52.89 | 0.692 | 65.41 | 0.791 | 60.49 | 0.754 | 55.37 | 0.713 | 58.33 | 0.737 | 58.81 | 0.700 |
| ★Qwen2.5-VL (32B) | 61.16 | 0.759 | 73.68 | 0.848 | 82.93 | 0.907 | 44.63 | 0.617 | 74.48 | 0.854 | 69.82 | 0.780 |
| ★InternVL2.5 (38B) | 55.37 | 0.713 | 68.42 | 0.813 | 70.73 | 0.829 | 43.80 | 0.609 | 64.06 | 0.781 | 62.05 | 0.732 |
| ★InternVL3 (38B) | 73.55 | 0.848 | 78.20 | 0.878 | 80.98 | 0.895 | 54.55 | 0.706 | 79.69 | 0.887 | 74.87 | 0.815 |
| ★Qwen2.5-VL (72B) | 61.16 | 0.759 | 75.94 | 0.863 | 76.10 | 0.864 | 41.32 | 0.585 | 68.75 | 0.815 | 66.45 | 0.760 |
| ★InternVL3 (78B) | 58.68 | 0.740 | 74.44 | 0.853 | 72.20 | 0.839 | 45.45 | 0.625 | 66.67 | 0.800 | 64.90 | 0.755 |
| △Gemini1.5-pro | 87.37 | 0.933 | 93.64 | 0.967 | 93.33 | 0.966 | 85.57 | 0.922 | 94.00 | 0.969 | 91.41 | 0.955 |
| △GPT4o | 83.16 | 0.908 | 90.91 | 0.952 | 92.12 | 0.959 | 73.20 | 0.845 | 89.33 | 0.944 | 86.87 | 0.930 |
| **Model Average (Zero-shot)** | 64.11 | 0.759 | 70.43 | 0.802 | 70.31 | 0.802 | 54.41 | 0.680 | 66.20 | 0.774 | 65.87 | 0.735 |
| ♦Swin3D_T* | 65.26 | 0.790 | 77.27 | 0.872 | 83.03 | 0.907 | 70.10 | 0.824 | 76.67 | 0.868 | 75.69 | 0.862 |
| ♦ResNet3D_18* | 100.0 | 1.000 | 99.09 | 0.995 | 98.79 | 0.994 | 97.94 | 0.990 | 98.67 | 0.993 | 98.87 | 0.994 |
| ♦InternVL2.5 (8B)* | 100.0 | 1.000 | 100.0 | 1.000 | 100.0 | 1.000 | 100.0 | 1.000 | 100.0 | 1.000 | 100.0 | 1.000 |
| ♦InternVL3 (9B)* | 100.0 | 1.000 | 100.0 | 1.000 | 100.0 | 1.000 | 100.0 | 1.000 | 100.0 | 1.000 | 100.0 | 1.000 |

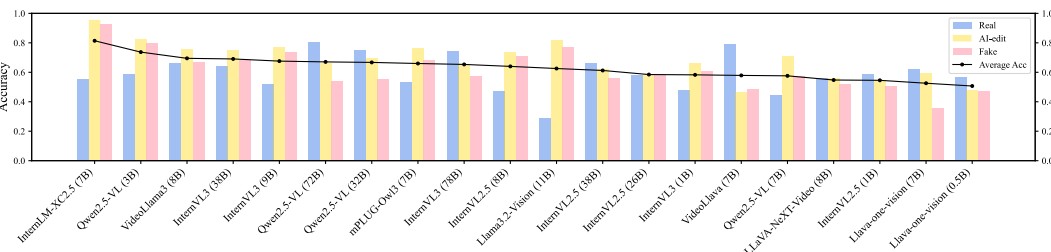

Figure 4: Deepfake video detection performance comparison of the open-source LMMs. We report zero-shot overall accuracy.

and AIGVDet (Bai et al., 2024), which are trained on specific deepfake datasets, show limited zero-shot generalization. The detection models trained on earlier deepfake datasets perform well on known fakes but struggle when exposed to new generative models that are not part of the training data. This limitation arises because these models often learn artifacts specific to the training dataset, leading to poor performance on unseen data. In contrast, LMMs, despite lacking task-specific training for real-fake discrimination, demonstrate relatively robust zero-shot detection performance. These models, such as InternLM-XComposer2.5 (7B), show impressive results even without fine-tuning specifically for deepfake detection tasks. However, once fine-tuned for the task, both traditional deep learning-based models and LMMs achieve 100% accuracy on detection tasks. Therefore, the ability to generalize in a zero-shot setting is more critical for deepfake detection, as unseen generative models are constantly evolving.

We also launch benchmarks on the real video subsets. From Table 3, we can observe that models exhibit variable performance across different datasets, excelling on more structured datasets such as LIVEVQC and LSVQ but struggling on LIVE-YT-Gaming. This highlights the sensitivity of some models to content type. DeMamba performs the best on real datasets but the worst on AI-generated datasets, indicating that it is biased towards real data and struggle to generalize to AI-generated content. Qwen2.5-VL (72B) and Gemini1.5-pro exhibit high zero-shot accuracy, demonstrating their strong ability to generalize across different real video datasets without any task-specific fine-tuning. Similarly, once fine-tuned for the specific task of deepfake detection, both traditional deep learning models and LMMs achieve near-perfect accuracy, with detection rates approaching 100%. While fine-tuning enhances the performance of traditional deep-learning-based models, LMMs offer greater flexibility, as they are able to perform well without specific task training.

Table 5: Results of cross-generator validation on different training and testing subsets using **Swin3D_T**.

| Testing Subset | Training Subset | | | | | | | | | | | | | | | | | | Avg Acc |
|---|---|---|---|---|---|---|---|---|---|---|---|---|---|---|---|---|---|---|---|
| | Alle | Anim | CogV | Hot | LTX | LVDM | LWM | Latt | Lavi | MS | NOVA | T2VZ | Pyra | TAV | VC1 | VC2 | Wan | ZS | |
| Allegro | **99.3** | 88.7 | 7.0 | 31.3 | 2.3 | 0.0 | 8.7 | 25.3 | 37.7 | 0.7 | 73.0 | 59.3 | 1.3 | 4.7 | 0.3 | 22.0 | 52.3 | 0.0 | 28.6 |
| Animate | 74.3 | **94.0** | 17.7 | 43.3 | 4.3 | 0.3 | 28.7 | 33.7 | 49.3 | 1.0 | 73.7 | 57.7 | 1.0 | 2.3 | 1.7 | 17.3 | 48.0 | 0.3 | 30.5 |
| CogVideo | 54.0 | 68.7 | **96.7** | 68.7 | 13.7 | 0.0 | 13.3 | 30.3 | 69.0 | 0.7 | 72.0 | 45.3 | 0.0 | 0.0 | 4.7 | 19.0 | 53.7 | 3.0 | 34.0 |
| Hotshot-XL | 32.3 | 42.0 | 13.7 | **98.3** | 5.0 | 0.3 | 20.0 | 37.7 | 27.0 | 5.7 | 86.0 | 24.7 | 6.7 | 5.3 | 11.0 | 53.0 | 27.7 | 19.7 | 28.7 |
| LTX | 34.0 | 44.3 | 6.7 | 44.7 | **99.7** | 0.0 | 17.3 | 30.3 | 42.3 | 1.7 | 66.0 | 28.0 | 0.7 | 11.3 | 6.3 | 22.7 | 44.7 | 2.3 | 27.9 |
| LVDM | 16.0 | 15.0 | 0.7 | 8.0 | 0.0 | **99.7** | 89.3 | 33.3 | 43.7 | 98.7 | 21.0 | 8.0 | 13.3 | 20.3 | 50.0 | 44.3 | 15.7 | 0.0 | 32.1 |
| LWM | 29.7 | 35.3 | 2.3 | 38.0 | 1.7 | 1.0 | **99.3** | 41.3 | 22.3 | 18.0 | 60.0 | 19.7 | 4.7 | 8.7 | 3.0 | 25.3 | 19.3 | 0.7 | 23.9 |
| Latte | 62.3 | 63.3 | 12.7 | 42.0 | 6.3 | 1.7 | 56.0 | **81.3** | 63.7 | 2.0 | 73.0 | 41.7 | 6.0 | 17.7 | 4.7 | 47.0 | 52.0 | 2.0 | 35.3 |
| Lavie | 58.0 | 66.7 | 14.0 | 49.0 | 8.0 | 1.0 | 26.0 | 44.7 | **92.7** | 1.7 | 74.3 | 45.7 | 6.7 | 8.0 | 1.3 | 43.0 | 58.0 | 1.7 | 33.4 |
| ModelScope | 7.7 | 17.0 | 0.7 | 50.7 | 0.0 | 75.7 | 95.0 | 59.3 | 14.3 | **99.0** | 47.0 | 1.7 | 2.3 | 28.0 | 84.7 | 54.3 | 11.0 | 12.3 | 36.7 |
| NOVA | 70.0 | 80.7 | 22.3 | 87.0 | 4.3 | 0.0 | 21.7 | 39.3 | 50.0 | 0.3 | **97.7** | 57.0 | 1.3 | 2.0 | 2.7 | 43.3 | 51.7 | 4.3 | 35.3 |
| Pyramid | 96.0 | 86.3 | 15.7 | 46.7 | 5.3 | 0.3 | 22.7 | 38.7 | 80.3 | 1.3 | 85.3 | **98.7** | 5.3 | 8.0 | 1.0 | 63.3 | 70.0 | 0.7 | 40.3 |
| T2V-Zero | 44.3 | 32.0 | 0.0 | 10.3 | 0.7 | 0.0 | 46.3 | 53.3 | 75.0 | 0.3 | 40.0 | 36.7 | **99.3** | 38.7 | 1.7 | 71.7 | 10.0 | 0.3 | 31.1 |
| Tune-A-Video | 19.7 | 18.7 | 0.3 | 3.3 | 1.0 | 0.0 | 24.0 | 45.3 | 26.3 | 1.7 | 8.3 | 9.0 | 6.0 | **88.7** | 5.0 | 30.0 | 20.3 | 2.3 | 17.2 |
| VideoCrafer1 | 15.7 | 31.0 | 0.7 | 69.0 | 0.0 | 55.0 | 19.0 | 30.3 | 24.0 | 86.3 | 49.3 | 7.3 | 2.0 | 22.7 | **99.7** | 75.3 | 18.7 | 22.7 | 34.9 |
| VideoCrafer2 | 73.0 | 72.0 | 8.0 | 77.0 | 2.3 | 1.7 | 37.7 | 37.7 | 78.7 | 2.0 | 84.7 | 60.3 | 11.3 | 28.7 | 10.7 | **99.3** | 66.7 | 5.3 | 42.1 |
| Wan2.1 | 79.3 | 78.7 | 12.3 | 38.3 | 2.0 | 0.0 | 8.7 | 38.7 | 81.0 | 1.3 | 73.3 | 61.0 | 0.3 | 4.7 | 4.3 | 44.7 | **96.3** | 0.0 | 34.7 |
| ZeroScope | 18.0 | 43.7 | 0.7 | 84.0 | 1.0 | 0.0 | 12.7 | 18.7 | 13.3 | 3.0 | 51.0 | 21.3 | 7.3 | 27.3 | 57.3 | 75.3 | 13.0 | **92.0** | 30.0 |
| ChatGLM | 43.3 | 50.7 | 11.7 | 28.0 | 2.3 | 0.7 | 3.0 | 9.3 | 74.3 | 0.7 | 60.3 | 35.3 | 5.3 | 3.0 | 8.3 | 19.7 | 35.0 | 0.7 | 21.8 |
| Gen3 | 56.0 | 61.4 | 1.6 | 22.5 | 0.6 | 0.0 | 11.0 | 22.5 | 60.6 | 0.0 | 54.2 | 41.8 | 5.4 | 11.6 | 1.6 | 29.7 | 36.7 | 1.2 | 23.2 |
| Genmo | 78.3 | 55.3 | 2.0 | 55.7 | 0.7 | 0.0 | 6.0 | 24.3 | 38.7 | 2.3 | 73.7 | 67.0 | 12.3 | 4.3 | 5.7 | 50.7 | 41.7 | 4.3 | 29.1 |
| Hunyuan | 47.7 | 34.0 | 3.7 | 34.0 | 0.3 | 0.0 | 4.3 | 6.0 | 31.3 | 2.3 | 47.7 | 35.0 | 3.0 | 5.0 | 3.0 | 28.7 | 43.0 | 0.0 | 18.3 |
| Hailuo | 42.7 | 30.0 | 2.0 | 15.0 | 0.3 | 0.0 | 1.0 | 4.7 | 31.0 | 0.7 | 25.7 | 30.7 | 3.0 | 7.7 | 0.3 | 17.0 | 38.3 | 0.3 | 13.9 |
| Jimeng | 38.7 | 28.7 | 2.7 | 29.3 | 0.3 | 0.3 | 0.3 | 2.3 | 22.3 | 0.3 | 28.0 | 25.7 | 2.7 | 3.3 | 3.7 | 19.7 | 29.0 | 0.0 | 13.2 |
| Kling | 37.0 | 35.3 | 3.3 | 44.0 | 2.0 | 0.0 | 4.7 | 6.3 | 16.7 | 1.3 | 60.7 | 30.3 | 3.7 | 10.3 | 5.3 | 34.0 | 31.3 | 12.3 | 18.8 |
| Pixverse | 57.7 | 55.3 | 3.0 | 49.3 | 1.7 | 0.7 | 11.3 | 9.3 | 33.0 | 1.3 | 58.7 | 46.3 | 2.3 | 5.0 | 3.3 | 46.0 | 35.7 | 1.7 | 23.4 |
| Sora | 40.0 | 40.3 | 2.7 | 25.0 | 0.7 | 0.0 | 1.0 | 10.0 | 30.0 | 1.0 | 29.7 | 25.3 | 3.3 | 13.7 | 4.3 | 28.3 | 37.7 | 1.7 | 16.4 |
| Vidu1.5 | 62.0 | 48.0 | 2.7 | 10.3 | 0.0 | 1.0 | 9.0 | 6.3 | 42.7 | 2.3 | 37.7 | 42.3 | 5.0 | 3.0 | 4.3 | 35.3 | 41.0 | 0.0 | 19.6 |
| Wanxiang | 62.7 | 57.0 | 3.3 | 42.7 | 3.7 | 0.7 | 3.3 | 13.7 | 25.3 | 1.3 | 61.3 | 46.3 | 3.3 | 5.7 | 1.7 | 25.7 | 45.0 | 0.0 | 22.4 |
| Xunfei | 66.0 | 58.0 | 8.3 | 60.3 | 0.7 | 0.7 | 6.3 | 12.7 | 13.0 | 2.0 | 65.7 | 50.3 | 0.3 | 1.7 | 1.3 | 30.7 | 36.3 | 3.7 | 23.2 |
| Avg Acc | 50.5 | 51.1 | 9.3 | 43.5 | 5.7 | 8.0 | 23.6 | 28.2 | 43.7 | 11.4 | **58.0** | 38.6 | 7.5 | 13.4 | 13.1 | 40.5 | 39.3 | 6.5 | 27.3 |

Table 6: Results of cross-generator validation on different training and testing subsets using **InternVL2.5 (8B)**.

| Testing Subset | Training Subset | | | | | | | | | | | | | | | | | | Avg Acc |
|---|---|---|---|---|---|---|---|---|---|---|---|---|---|---|---|---|---|---|---|
| | Alle | Anim | CogV | Hot | LTX | LVDM | LWM | Latt | Lavi | MS | NOVA | T2VZ | Pyra | TAV | VC1 | VC2 | Wan | ZS | |
| Allegro | **100.0** | 58.9 | 44.3 | 4.7 | 66.3 | 0.0 | 0.0 | 24.3 | 8.0 | 0.0 | 90.7 | 83.3 | 30.1 | 23.1 | 4.7 | 12.7 | 96.6 | 19.7 | 37.1 |
| Animate | 46.7 | **100.0** | 34.0 | 55.7 | 74.7 | 0.3 | 7.3 | 33.7 | 24.7 | 0.0 | 44.0 | 30.3 | 3.6 | 3.4 | 15.3 | 52.8 | 58.8 | 26.3 | 34.0 |
| CogVideo | 53.3 | 45.6 | **100.0** | 10.0 | 52.0 | 0.3 | 1.3 | 25.0 | 35.0 | 0.0 | 47.3 | 42.3 | 14.4 | 10.9 | 34.0 | 25.0 | 74.7 | 40.8 | 34.0 |
| Hotshot-XL | 52.7 | 58.6 | 67.0 | **100.0** | 95.7 | 0.0 | 7.7 | 64.0 | 17.7 | 0.3 | 34.0 | 25.0 | 17.7 | 24.8 | 76.3 | 65.2 | 80.1 | 92.8 | 48.9 |
| LTX | 44.7 | 77.8 | 26.0 | 18.3 | **100.0** | 0.0 | 1.3 | 60.7 | 4.0 | 0.0 | 41.3 | 27.7 | 3.6 | 24.3 | 53.0 | 51.5 | 64.7 | 67.1 | 37.0 |
| LVDM | 0.0 | 3.0 | 0.0 | 0.0 | 2.3 | **100.0** | 100.0 | 93.0 | 0.3 | 100.0 | 0.0 | 0.0 | 82.0 | 27.7 | 7.0 | 79.7 | 1.1 | 10.8 | 33.7 |
| LWM | 3.0 | 28.3 | 1.3 | 0.0 | 10.0 | 6.7 | **100.0** | 98.7 | 3.7 | 38.0 | 6.7 | 5.3 | 17.6 | 50.0 | 16.7 | 30.9 | 7.2 | 19.0 | 24.6 |
| Latte | 37.0 | 69.0 | 1.3 | 1.3 | 57.0 | 0.0 | 18.3 | **100.0** | 56.0 | 0.3 | 42.0 | 27.3 | 49.8 | 83.5 | 22.7 | 71.2 | 68.9 | 37.6 | 43.5 |
| Lavie | 24.7 | 70.1 | 50.0 | 1.7 | 20.0 | 0.7 | 0.7 | 86.0 | **100.0** | 0.0 | 47.3 | 43.7 | 23.7 | 14.3 | 28.0 | 75.0 | 74.7 | 40.1 | 38.9 |
| ModelScope | 2.0 | 17.6 | 4.0 | 5.7 | 37.7 | 88.7 | 100.0 | 91.7 | 0.3 | **100.0** | 0.0 | 0.0 | 22.3 | 32.1 | 32.3 | 60.4 | 8.8 | 61.9 | 37.0 |
| NOVA | 70.7 | 70.5 | 63.0 | 15.3 | 86.7 | 0.0 | 0.0 | 30.7 | 13.7 | 0.0 | **100.0** | 40.7 | 3.3 | 12.7 | 32.7 | 52.9 | 79.7 | 52.0 | 37.9 |
| Pyramid | 100.0 | 20.2 | 72.7 | 3.3 | 31.7 | 0.0 | 0.0 | 9.0 | 31.3 | 0.0 | 100.0 | 68.9 | **100.0** | 41.9 | 9.0 | 3.9 | 99.9 | 18.5 | 39.4 |
| T2V-Zero | 75.7 | 35.3 | 18.0 | 2.0 | 8.7 | 1.0 | 1.3 | 100.0 | 89.3 | 0.0 | 81.7 | 79.0 | **100.0** | 100.0 | 9.7 | 49.3 | 92.0 | 29.6 | 48.5 |
| Tune-A-Video | 28.0 | 8.6 | 1.0 | 1.0 | 8.3 | 1.0 | 6.3 | 99.3 | 38.0 | 1.0 | 30.7 | 21.7 | 62.3 | **100.0** | 15.3 | 53.9 | 71.9 | 38.2 | 32.6 |
| VideoCrafer1 | 0.0 | 36.7 | 7.7 | 4.7 | 53.0 | 77.7 | 0.7 | 52.7 | 3.3 | 61.7 | 0.0 | 0.0 | 0.0 | 1.3 | **100.0** | 96.0 | 0.7 | 85.9 | 30.5 |
| VideoCrafer2 | 20.7 | 84.7 | 32.0 | 2.0 | 37.7 | 0.3 | 0.0 | 97.3 | 88.3 | 0.0 | 34.7 | 26.3 | 11.6 | 22.0 | 74.3 | **100.0** | 37.5 | 86.4 | 42.0 |
| Wan2.1 | 79.3 | 78.7 | 12.3 | 38.3 | 2.0 | 0.0 | 8.7 | 38.7 | 81.0 | 1.3 | 73.3 | 61.0 | 0.3 | 4.7 | 4.3 | 44.7 | **100.0** | 0.0 | 34.7 |
| ZeroScope | 7.3 | 61.0 | 26.0 | 29.0 | 78.3 | 0.0 | 1.3 | 11.3 | 9.0 | 3.0 | 3.3 | 2.0 | 0.8 | 10.0 | 100.0 | 68.7 | 14.2 | **100.0** | 29.2 |
| ChatGLM | 45.3 | 2.7 | 54.0 | 0.3 | 6.3 | 0.3 | 0.0 | 10.3 | 0.0 | 0.0 | 50.7 | 43.3 | 11.6 | 8.8 | 5.0 | 1.7 | 80.9 | 13.8 | 18.8 |
| Gen3 | 87.3 | 17.7 | 26.7 | 1.0 | 8.7 | 0.0 | 0.3 | 8.0 | 21.3 | 0.0 | 86.3 | 81.3 | 38.8 | 36.4 | 7.7 | 6.5 | 98.6 | 18.2 | 30.3 |
| Genmo | 89.0 | 39.6 | 63.3 | 1.7 | 22.3 | 0.0 | 0.0 | 13.0 | 17.3 | 0.0 | 92.3 | 86.7 | 36.3 | 23.4 | 12.7 | 21.0 | 99.1 | 44.7 | 36.4 |
| Hunyuan | 25.0 | 0.0 | 30.3 | 0.0 | 1.0 | 0.0 | 0.0 | 1.0 | 0.0 | 0.0 | 1.7 | 6.0 | 0.0 | 0.8 | 0.0 | 0.0 | 13.3 | 6.8 | 4.8 |
| Hailuo | 85.3 | 3.1 | 29.7 | 0.7 | 5.0 | 0.0 | 0.0 | 0.7 | 2.0 | 0.0 | 84.7 | 75.0 | 18.6 | 16.2 | 3.0 | 0.0 | 97.5 | 14.6 | 24.2 |
| Jimeng | 42.3 | 0.0 | 30.3 | 0.0 | 0.7 | 0.0 | 0.0 | 0.7 | 0.3 | 0.0 | 26.7 | 31.0 | 0.4 | 13.1 | 0.0 | 0.0 | 65.9 | 6.9 | 12.1 |
| Kling | 42.0 | 2.8 | 35.0 | 0.7 | 10.0 | 0.0 | 0.0 | 1.0 | 0.0 | 0.0 | 39.0 | 30.3 | 0.8 | 2.9 | 9.7 | 7.8 | 64.1 | 21.3 | 14.9 |
| Pixverse | 52.3 | 70.8 | 26.0 | 5.0 | 52.7 | 0.0 | 0.0 | 6.0 | 14.0 | 0.0 | 60.0 | 38.3 | 1.6 | 5.0 | 39.7 | 73.8 | 62.0 | 71.4 | 32.1 |
| Sora | 78.0 | 13.6 | 23.0 | 1.7 | 12.7 | 0.0 | 0.0 | 3.7 | 6.3 | 0.0 | 74.7 | 64.0 | 26.5 | 22.5 | 6.0 | 8.6 | 98.9 | 18.9 | 25.5 |
| Vidu1.5 | 33.0 | 88.5 | 15.7 | 1.3 | 32.3 | 0.0 | 0.0 | 19.7 | 14.3 | 0.0 | 51.7 | 30.0 | 2.3 | 4.5 | 9.3 | 41.7 | 45.2 | 24.2 | 23.0 |
| Wanxiang | 29.0 | 2.9 | 38.7 | 0.0 | 1.7 | 0.0 | 0.0 | 0.7 | 0.0 | 0.0 | 3.3 | 7.0 | 0.4 | 1.6 | 0.0 | 2.4 | 8.3 | 19.2 | 6.4 |
| Xunfei | 82.0 | 48.8 | 49.7 | 2.0 | 25.3 | 0.0 | 0.0 | 25.7 | 4.3 | 0.0 | 75.7 | 60.7 | 4.9 | 14.0 | 15.3 | 52.1 | 76.9 | 60.9 | 33.2 |
| Avg Acc | 48.0 | 38.6 | 35.6 | 9.1 | 34.1 | 9.2 | 11.6 | 39.2 | 21.5 | 10.1 | 46.6 | 39.6 | 22.9 | 25.0 | 23.7 | 38.9 | **61.1** | 38.7 | 30.8 |

We further evaluate the performance of different detection models on AI-edit video subsets, as shown in Table 4. The AI-edit video subsets consist of five video-editing categories: background, object operation, style change, color change, and action edit, each posing different challenges for detection models. Among these, models achieve the highest average accuracy on style change and the lowest on the action edit category, which involves subtle modifications to the appearance of individual objects. These results suggest that subtle changes at the object level are more difficult for detection models to identify compared to style changes. Models show significant improvements after fine-tuning, especially LMMs outperform conventional deep-learning-based networks, highlighting the effectiveness of LMMs in deepfake detection tasks.

We summarize the overall in-domain performance across the three subsets in Figure 4. Models such as Qwen2.5-VL (7B) and VideoLlama3 (8B) consistently perform well on fully fake and AI-edited videos; however, their performance slightly declines on real videos. This suggests that while these models are effective at identifying manipulated content, they struggle more with real videos containing natural distortions or quality degradation. Furthermore, the average accuracy trend (black curve) indicates that although most models excel on real content, their overall accuracy decreases when evaluated on fake videos. This drop may be attributed to the unique and often sophisticated characteristics of deepfake content, which can deceive most detectors.

### 4.3 CROSS-GENERATOR VIDEO DETECTION

From Table 5 and Table 6, we evaluate the performance of deepfake detection models trained and evaluated across different video generation models. The detection model is trained on a specific generator and then tested on a variety of other generators. This setup allows us to assess how well the models generalize across different generators, which is crucial for real-world applications where deepfake detection systems may encounter novel generative techniques that are not part of the training data.

We train the Swin3D_T and Intern2.5 (8B) on 18 open-source video generation models and then evaluate them on the test set of 30 video generation models, including both open-source and closed-source generators. The results show that models exhibit nearly perfect accuracy close to 100% on generators they have seen during training, as indicated by the bolded diagonal values in the table. This indicates that these models are highly specialized in detecting fakes from the specific generator they are trained on. However, the performance significantly drops when the models are tested on generators they haven't encountered before, indicating that while detection models can perform exceptionally well on known generative models, their ability to generalize to new and unseen generative techniques remains limited. This underscores the importance of ensuring that deepfake detection systems can generalize effectively, especially as generative models continue to evolve rapidly. It highlights a key challenge for future research in improving the robustness and adaptability of these systems to more advanced video generation models.

Table 7: Results of cross-generator validation on 18 open-source generators' training and 12 unseen generators' testing subsets using **Swin3D_T** and **ResNet3D_18**. ♡Open-source lab T2V models. ♠Close-source commercial T2V models.

| Testing models | Training methods | | Avg Acc |
|---|---|---|---|
| | Swin3D_T | ResNet3D_18 | |
| ♡Allegro | 100% | 99.7% | 99.9% |
| ♡Animate | 98.7% | 96.0% | 97.4% |
| ♡CogVideo | 99.3% | 100% | 99.7% |
| ♡Hotshot-XL | 99.3% | 99.0% | 99.2% |
| ♡LTX | 99.7% | 98.3% | 99.0% |
| ♡LVDM | 100% | 100% | 100% |
| ♡LWM | 98.3% | 99.0% | 98.7% |
| ♡Latte | 100% | 100% | 100% |
| ♡Lavie | 99.3% | 99.0% | 99.2% |
| ♡ModelScope | 100% | 100% | 100% |
| ♡NOVA | 100% | 100% | 100% |
| ♡Pyramid | 100% | 99.7% | 99.9% |
| ♡T2V-Zero | 99.3% | 100% | 99.7% |
| ♡Tune-A-Video | 98.7% | 97.3% | 98.0% |
| ♡VideoCrafer1 | 100% | 100% | 100% |
| ♡VideoCrafer2 | 100% | 100% | 100% |
| ♡Wan2.1 | 99.0% | 100% | 99.5% |
| ♡ZeroScope | 100% | 100% | 100% |
| ♠ChatGLM | 98.0% | 98.3% | 98.2% |
| ♠Gen3 | 95.2% | 96.0% | 95.6% |
| ♠Genmo | 97.0% | 98.7% | 97.9% |
| ♠Hunyuan | 84.0% | 88.3% | 86.2% |
| ♠Hailuo | 87.3% | 89.3% | 88.3% |
| ♠Jimeng | 71.0% | 82.0% | 76.5% |
| ♠Kling | 87.0% | 89.0% | 88.0% |
| ♠Pixverse | 96.3% | 97.7% | 97.0% |
| ♠Sora | 91.7% | 92.3% | 92.0% |
| ♠Vidu1.5 | 90.0% | 91.0% | 90.5% |
| ♠Wanxiang | 93.7% | 96.0% | 94.9% |
| ♠Xunfei | 97.3% | 85.0% | 91.2% |
| Avg Acc | 96.0% | 96.4% | 96.2% |

To further evaluate the generalization ability of our models across diverse generative sources, we train models on a dataset composed of 18 open-source video generation models and tested them on a held-out set of 12 unseen commercial (closed-source) generators. This setting enables a rigorous cross-generator validation, reflecting the model's ability to generalize to previously unseen generative distributions. As shown in Table 7, both Swin3D_T and ResNet3D_18 achieve near-perfect accuracy on the seen open-source generators, indicating strong discriminative capability within the training domain. However, their performance degrades to a certain extent on the unseen commercial generators. These results underscore the importance of model scalability and architectural robustness in cross-domain generalization, particularly when handling distribution shifts between training and testing generators.

## 5 CONCLUSION

In this paper, we present FVBench, a comprehensive benchmark designed to overcome the limitations of existing datasets for deepfake video detection. FVBench consists of 120K videos spanning real, AI-edited, and fully AI-generated content, with an emphasis on enhancing both content variety and generative model diversity. We conduct one of the first in-depth studies exploring the potential of LMMs in deepfake video detection. Our results highlight that, while fine-tuned deepfake detection models excel at detecting known fakes, their performance significantly drops when confronted with previously unseen generation models. This underscores the importance of zero-shot generalization in future detection systems, which is crucial as generative contents continue to become more realistic. We hope FVBench serves as a catalyst for the development of next-generation detection methods and inspires further research into scalable and adaptive content authenticity solutions.

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

## A  APPENDIX

We clarify the use of LLM in section B, the detail information of T2V models in section C, detail information of the LMMs in section D, more result comparisons in section E and the training hyperparameters in section F.

## B  USE OF LLM

During the drafting of this manuscript, LLM (GPT4o) was employed to enhance the text's fluency, rectify grammatical inaccuracies, and improve the precision of phrasing. However, the central ideas, experimental design, and final conclusions are generated without AI contribution.

## C  DETAILED INFORMATION OF T2V MODELS

**Pyramid** Jin et al. (2024) introduces a novel pyramidal flow matching strategy for video generation, which progressively aligns data distributions at multiple scales. This enables high-quality synthesis while potentially reducing computational demand.

**Wan2.1** Wang et al. (2025a) is an open-source large-scale video generation model developed under the "Wan" initiative. It is optimized for high-fidelity video creation and aims to enhance accessibility and transparency in advanced generative AI research.

**Allegro** Zhou et al. (2024) aims to replicate commercial-grade video synthesis within a transparent framework. Its goal is to demystify the "black box" nature of high-end video models while striving for comparable visual quality.

**VideoCrafter2** Chen et al. (2024a) addresses the lack of high-quality video training data by separating motion and appearance at the data level. It utilizes low-resolution videos for learning motion and high-quality images for preserving appearance.

**CogVideo X1.5** Yang et al. (2024b) is a diffusion-based text-to-video model featuring an Expert Transformer, designed to improve efficiency, scalability, and specialization in complex video synthesis.

**Animate** Xu et al. (2024) proposes a transformer-based method for generating long-form videos with temporal consistency, focusing on modeling long-range dependencies to preserve narrative coherence.

**Lavie** Wang et al. (2023b) adopts a cascaded latent diffusion architecture, generating and refining video content in stages within the latent space. This design improves detail and coherence in high-resolution video generation.

**Hotshot-XL** Mullan et al. (2023) is a text-to-GIF model adapted for video output by converting GIFs to MP4 format. It is built on Stable Diffusion XL and employs default settings from its official implementation.

**Latte** Ma et al. (2025) is a latent diffusion video model based on a Transformer backbone. It encodes videos via a pretrained VAE and processes spatial-temporal tokens using four architectural variants for efficient and expressive synthesis.

**VideoCrafter1** Chen et al. (2023) presents two diffusion models: one for text-to-video and another for image-to-video generation. By incorporating temporal attention, it enhances consistency in videos generated from large-scale datasets.

**NOVA** Deng et al. (2024) reframes video generation as an autoregressive prediction task across time and space. This design allows efficient frame-by-frame synthesis and supports flexible, zero-shot generative capabilities.

**ModelScope** Wang et al. (2023a) proposes a decomposed diffusion approach that separates base and residual noise across frames. This improves spatial-temporal consistency while benefiting from pretrained generative modules.

**Text2Video-Zero** Khachatryan et al. (2023) is a zero-shot text-to-video model that requires no fine-tuning. It introduces motion dynamics between latent codes and uses cross-frame attention to ensure temporal coherence.

**Tune-A-Video** Wu et al. (2023) is a one-shot text-to-video generation model employing sparse spatio-temporal attention. It can generate coherent videos from a single example and supports conditional or personalized inputs.

**LTX** HaCohen et al. (2024) is a real-time video generation model based on latent diffusion. Its key strength lies in reducing generation latency, allowing interactive video synthesis without compromising quality.

**LVDM** He et al. (2022) leverages a hierarchical diffusion design in latent space to extend generation length and reduce computational cost. It includes conditional perturbation and guidance to maintain quality across extended durations.

**ZeroScope** Team (2024h) serves as a specialized upscaler for outputs generated with zeroscope_v2_576w. By converting low-resolution previews to high-resolution videos, it enables faster iteration with high visual fidelity.

**LWM** Liu et al. (2024a) is a multimodal autoregressive model that uses RingAttention to handle long sequences efficiently. It supports strong video-language understanding and generation with a context window of up to 1M tokens.

**Pixverse** AI (2024) is an all-in-one AI video tool that supports viral effects, video-to-video restyling, lip-sync, and AI-based video extension. It's beginner-friendly yet powerful enough for advanced users, making it ideal for both casual and professional video creation.

**Wanxiang** Cloud (2024), developed by Alibaba DAMO Academy, is a multimodal large model capable of cross-modal generation and understanding across text, video, and audio. It supports tasks such as text-to-video and visual question answering.

**Hailuo** Team (2024d), by MiniMax, enables text-to-video and video editing through simple prompts. It offers an accessible platform for generating high-quality videos in marketing, education, and entertainment.

**Jimeng** Team (2024a), from Faceu Technology, is a text-to-video model focused on generating short, realistic video clips with precise prompt interpretation.

**Sora** Team (2024e) excels in prompt understanding, generating emotionally expressive characters, multi-shot videos, and complex scenes with consistent motion and detail.

**Hunyuan** Li et al. (2024c) is a 13B-parameter open-source text-to-video model that produces videos with strong physical realism and scene consistency, supporting creative visual generation.

**Vidu 1.5** Team (2024f) introduces Multiple-Entity Consistency, allowing unrelated characters, objects, and environments to be seamlessly combined into visually coherent videos, even with complex inputs.

**Gen3** Runway (2024) marks a new generation of foundation models from Runway, built on a fresh large-scale multimodal training infrastructure. Trained jointly on video data, Gen-3 Alpha powers a range of tools including text-to-video, video-to-video, and motion editing modes like motion brush and director mode, while also supporting upcoming features for finer control over structure, style, and motion.

**Kling** Team (2024c), developed by Kuaishou's Large Model Algorithm Team, represents a new class of AI creativity tools, offering rich capabilities for generating and editing AI-generated video content with high controllability.

**Genmo** Team (2024b) is an AI creation assistant designed for video generation and editing. Users can create animations and stylized videos from text or images, as well as restyle existing footage, making it a versatile platform for creative exploration.

**ChatGLM** GLM et al. (2024), from Zhipu AI and Tsinghua University's KEG lab, is a bilingual large language model family ranging from GLM-130B to the advanced GLM-4. The latest ver-

Table 8: An overview and URLs of the adopted 30 T2V generation models. ♡Open-source lab T2V models. ♠Close-source commercial T2V models. †Representative variable and optional.

| Models | Frames | FPS | Resolution | URL |
|---|---|---|---|---|
| ♡Pyramid Jin et al. (2024) | 121 | 24 | 1280×768 | https://github.com/jy0205/Pyramid-Flow |
| ♡Wan2.1 Wang et al. (2025a) | 81† | 16† | 832×480† | https://github.com/FoundationVision/LlamaGen |
| ♡Allegro Zhou et al. (2024) | 88 | 15 | 1280×720 | https://github.com/rhymes-ai/Allegro |
| ♡VideoCrafter2 Chen et al. (2024a) | 16 | 10 | 512×320 | https://github.com/AILab-CVC/VideoCrafter |
| ♡CogVideo X1.5 Yang et al. (2024b) | 32 | 8 | 1360×768 | https://github.com/THUDM/CogVideo |
| ♡Animate Xu et al. (2024) | 49 | 8 | 672×384 | https://github.com/aigc-apps/EasyAnimate |
| ♡Lavie Wang et al. (2023b) | 16 | 8 | 512×320 | https://github.com/Vchitect/LaVie |
| ♡Hotshot-XL Mullan et al. (2023) | 8 | 8 | 672×384 | https://github.com/hotshotco/Hotshot-XL |
| ♡Latte Ma et al. (2025) | 16 | 8 | 512×512 | https://github.com/Vchitect/Latte |
| ♡VideoCrafter1 Chen et al. (2023) | 16† | 10† | 512×320† | https://github.com/AILab-CVC/VideoCrafter |
| ♡Text2Video-Zero Khachatryan et al. (2023) | 8 | 4 | 512×512 | https://github.com/Picsart-AI-Research/Text2Video-Zero |
| ♡NOVA Deng et al. (2024) | 33 | 12 | 768×480 | https://github.com/baaivision/NOVA |
| ♡ModelScope Wang et al. (2023a) | 16 | 8 | 256×256 | https://github.com/modelscope/modelscope |
| ♡Tune-A-Video Wu et al. (2023) | 8 | 8 | 512×512 | https://github.com/showlab/Tune-A-Video |
| ♡LTX HaCohen et al. (2024) | 121 | 25 | 704×480 | https://github.com/Lightricks/LTX-Video |
| ♡LVDM He et al. (2022) | 16 | 8 | 256×256 | https://github.com/YingqingHe/LVDM |
| ♡ZeroScope Team (2024h) | 36 | 8 | 576×320 | https://huggingface.co/cerspense/zeroscope_v2_XL |
| ♡LWM Liu et al. (2024a) | 8 | 4 | 256×256 | https://github.com/LargeWorldModel/LWM |
| ♠Pixverse AI (2024) | 161† | 30† | 640×360† | https://pixverse.ai/ |
| ♠Wanxiang Cloud (2024) | 161† | 30† | 1280×720† | https://tongyi.aliyun.com/wanxiang/ |
| ♠Hailuo Team (2024d) | 141† | 25† | 1280×720† | https://hailuoai.video/ |
| ♠Jimeng Team (2024a) | 12†1 | 24† | 1472×832† | https://jimeng.jianying.com/ |
| ♠Sora Team (2024e) | 150† | 30† | 854×480† | https://openai.com/research/video-generation-models-as-world-simulators |
| ♠Hunyuan Li et al. (2024c) | 129† | 24† | 1280×720† | https://aivideo.hunyuan.tencent.com/ |
| ♠Vidu1.5 Team (2024f) | 60† | 16† | 688×384† | https://www.vidu.studio/zh |
| ♠Gen3 Runway (2024) | 128† | 24† | 1280×768† | https://runwayml.com/research/introducing-gen-3-alpha |
| ♠Kling Team (2024c) | 153† | 30† | 1280×720† | https://klingai.io/ |
| ♠Genmo Team (2024b) | 60† | 15† | 1728×1728† | https://www.genmo.ai |
| ♠ChatGLM GLM et al. (2024) | 151† | 30† | 1280×720† | https://chatglm.cn/video?lang=zh |
| ♠Xunfei Team (2024g) | 145† | 24† | 1024×576† | https://typemovie.art/ |

sion integrates an "All Tools" framework, enabling enhanced interaction with external modules for complex tasks.

**Xunfei** Team (2024g), by iFlytek, offers an AI-driven platform for quickly turning text into video. It simplifies video creation by providing a variety of styles and templates suited for producing short-form visual content efficiently.

# D DETAILED INFORMATION OF THE LMMS

**LLaVA-NeXT-Video** Li et al. (2024b) boosts video input resolution and enhances fine-grained perception capabilities, including OCR, visual reasoning, and factual knowledge grounding. It retains a compact training setup, relying on fewer than one million instruction-tuning samples to achieve high efficiency and broad generalization.

**VideoLLaVA** Liu et al. (2024b) presents a unified framework that bridges visual input with language representations by aligning visual features before projecting them into the language space. This approach empowers large language models to jointly reason over both images and video content within a shared architecture.

**InternVL2.5** Chen et al. (2024c) demonstrates strong multimodal capabilities across benchmarks involving cross-domain reasoning, document comprehension, and video analysis. It benefits from enhanced vision encoders, a larger training corpus, and optimized inference strategies, resulting in improved generalization and hallucination mitigation.

**InternVL3** Chen et al. (2024c) pushes the boundaries of multimodal LLMs by supporting a wider range of applications, including tool use, GUI interaction, industrial visual tasks, and 3D scene understanding. By unifying vision-language learning into a single-stage framework, it eliminates the need for additional adapters or fusion modules, streamlining training and improving scalability.

**VideoLlama3** Yao et al. (2024) adopts a four-stage training pipeline for vision-language modeling. It introduces innovations such as Rotary Position Embedding (RoPE) for adaptive image resolution handling and video token compression for efficient temporal representation, yielding strong performance across visual understanding tasks in both image and video modalities.

Table 9: **Performance benchmark on AI-generated video subsets as a supplement to the Table 2 in the main paper.** ♦* refers to finetuned models.

| Methods / Datasets | Genmo | Hailuo | T2V-Zero | Tune-A-Video | LVDM | LWM | LTX | ZeroScope | Jimeng | VCrafter2 |
|---|---|---|---|---|---|---|---|---|---|---|
| ♦AIGVDet* | 100.0% | 100.0% | 100.0% | 100.0% | 96.43% | 93.12% | 100.0% | 100.0% | 100.0% | 100.0% |
| ♦MC3_18* | 100.0% | 100.0% | 100.0% | 100.0% | 100.0% | 100.0% | 100.0% | 100.0% | 100.0% | 100.0% |

| Methods / Datasets | Xunfei | Hotshot | Hunyuan | CogVideo | NOVA | VCrafter1 | Sora | ChatGLM | Wan2.1 | Animate |
|---|---|---|---|---|---|---|---|---|---|---|
| ♦AIGVDet* | 100.0% | 100.0% | 100.0% | 100.0% | 100.0% | 100.0% | 100.0% | 100.0% | 100.0% | 100.0% |
| ♦MC3_18* | 100.0% | 100.0% | 100.0% | 100.0% | 100.0% | 100.0% | 100.0% | 100.0% | 100.0% | 100.0% |

| Methods / Datasets | Wanxiang | Allegro | Pyramid | Vidu1.5 | Lavie | Kling | Pixverse | Latte | MScope | Gen3 | Overall |
|---|---|---|---|---|---|---|---|---|---|---|---|
| ♦AIGVDet* | 100.0% | 100.0% | 98.24% | 100.0% | 100.0% | 100.0% | 100.0% | 100.0% | 100.0% | 100.0% | 99.59% |
| ♦MC3_18* | 100.0% | 100.0% | 100.0% | 100.0% | 100.0% | 100.0% | 100.0% | 100.0% | 100.0% | 100.0% | 100.0% |

Table 10: **Performance benchmark on real video subsets as a supplement to the Table 3 in the main paper.** ♡refers to model zero-shot results. ♦* refers to finetuned models.

| Methods / Datasets | MSRVTT | KoNViD | FineVD | WebVid | LSVQ | LIVEVQC | YouTubeUGC | LIVE-YT-Gaming | Overall |
|---|---|---|---|---|---|---|---|---|---|
| ♡MM-Det | 0.20% | 2.08% | 3.59% | 4.51% | 7.25% | 13.68% | 0.44% | 0.0% | 3.97% |
| ♦AIGVDet* | 99.43% | 97.81% | 98.32% | 100.0% | 96.89% | 99.42% | 100.0% | 99.17% | 98.88% |
| ♦MC3_18* | 99.75% | 99.48% | 98.77% | 99.67% | 97.56% | 97.85% | 100.0% | 100.0% | 98.91% |

**LLaMA3.2-Vision** Meta (2024) excels in video-based reasoning tasks, including understanding complex documents, interpreting data visualizations, and performing visual grounding. The model is capable of interpreting structured content such as charts and maps, while generating descriptive and context-aware captions for visual inputs.

**mPLUG-Owl3** Ye et al. (2024) is a robust multimodal model designed for understanding extended video sequences and interleaved video-text content. It features a novel Hyper Attention mechanism that fuses visual and textual signals into a shared embedding space, enabling effective processing of long-form and multi-video inputs.

**Qwen2.5-VL** Bai et al. (2025) represents the latest evolution of the Qwen vision-language family. It enhances recognition and localization capabilities, supports document-level reasoning, and improves long-video understanding through dynamic resolution scaling, absolute temporal encoding, and optimized inference via window-based attention mechanisms.

**LLaVA-One-Vision** Li et al. (2024a) is an open-source multimodal model designed for scalable visual-language learning across single images, image sequences, and video data. It features a cost-efficient architecture that links vision encoders with language models, enabling effective knowledge transfer from image to video tasks.

**InternLM-XComposer-2.5** Zhang et al. (2024b) is a powerful vision-language model built on InternLM2-7B, supporting long-context inputs up to 96K tokens. It also enables webpage generation and text-image article composition. It offers a strong open-source alternative for both vision-language understanding and content generation.

# E  MORE RESULT COMPARISONS

To further support the results presented in the main paper, we extend the experiments in the original tables by including additional models for training and evaluation, as shown in Tables 9- 12.

# F  TRAINING HYPERPARAMETERS

The following hyperparameters were consistently used across the training stages:

- **Dataset Split:** The dataset was partitioned into training and testing sets using a 4:1 ratio.
- **Learning Rate:** $4 \times 10^{-5}$ (*i.e.*, 4e−5).
- **Batch Size:** 4.
- **LoRA Configuration:**

Table 11: **Performance benchmark on AI-edit video subsets, including five editing types as a supplement to the Table 4 in the main paper.** ♡refers to model zero-shot results. ♦* refers to finetuned models.

| Dimension | Background | | Style Change | | Color Change | | Action Edit | | Object Operation | | Overall | |
|---|---|---|---|---|---|---|---|---|---|---|---|---|
| Methods / Metrics | Acc(%)↑ | F1↑ | Acc(%)↑ | F1↑ | Acc(%)↑ | F1↑ | Acc(%)↑ | F1↑ | Acc(%)↑ | F1↑ | Acc(%)↑ | F1↑ |
| ♡MM-Det | 95.10 | 0.935 | 100.00 | 1.000 | 97.67 | 0.947 | 96.94 | 0.960 | 98.82 | 1.000 | 97.85 | 0.837 |
| ♦AIGVDet* | 100.0 | 1.000 | 100.0 | 1.000 | 100.0 | 1.000 | 100.0 | 1.000 | 100.0 | 1.000 | 100.0 | 1.000 |
| ♦MC3_18* | 87.37 | 0.933 | 95.45 | 0.977 | 96.36 | 0.981 | 92.78 | 0.963 | 92.67 | 0.962 | 93.35 | 0.966 |

Table 12: Results of cross-generator validation on different training and testing subsets using **ResNet3D_18**.

| Testing Subset | Alle | Anim | CogV | Hot | LTX | LVDM | LWM | Latt | Lavi | MS | NOVA | T2VZ | Pyra | TAV | VC1 | VC2 | Wan | ZS | Avg Acc |
|---|---|---|---|---|---|---|---|---|---|---|---|---|---|---|---|---|---|---|---|
| Allegro | **99.3** | 83.7 | 3.0 | 17.0 | 23.0 | 0.0 | 13.0 | 46.7 | 44.3 | 0.3 | 33.3 | 79.0 | 1.0 | 5.7 | 0.0 | 34.0 | 81.7 | 3.0 | 31.6 |
| Animate | 49.3 | **94.7** | 13.3 | 26.3 | 30.0 | 0.0 | 34.7 | 56.3 | 47.0 | 1.7 | 32.0 | 60.7 | 0.3 | 4.7 | 2.0 | 19.0 | 61.0 | 4.3 | 29.9 |
| CogVideo | 49.7 | 63.3 | **99.3** | 40.0 | 47.7 | 0.0 | 25.7 | 59.7 | 66.0 | 0.3 | 48.0 | 40.3 | 0.0 | 1.7 | 1.0 | 13.7 | 42.7 | 3.3 | 33.5 |
| Hotshot-XL | 21.7 | 47.0 | 3.7 | **98.3** | 46.7 | 5.7 | 54.0 | 73.7 | 72.0 | 25.0 | 67.0 | 18.0 | 3.7 | 4.7 | 24.7 | 46.7 | 18.0 | 46.0 | 37.6 |
| LTX | 24.0 | 43.7 | 4.3 | 25.7 | **98.7** | 0.3 | 38.0 | 60.0 | 34.7 | 7.3 | 20.0 | 27.3 | 0.7 | 14.0 | 2.3 | 12.3 | 30.0 | 4.3 | 24.9 |
| LVDM | 24.0 | 5.7 | 0.0 | 14.3 | 1.3 | **100** | 92.0 | 83.7 | 65.0 | 98.7 | 11.3 | 25.7 | 30.7 | 24.7 | 88.3 | 61.0 | 31.7 | 1.3 | 42.2 |
| LWM | 5.7 | 14.0 | 0.7 | 23.3 | 12.3 | 18.3 | **99.7** | 64.0 | 15.3 | 28.0 | 13.3 | 14.0 | 9.7 | 23.7 | 10.3 | 17.7 | 13.0 | 1.7 | 21.4 |
| Latte | 41.3 | 61.3 | 8.7 | 27.3 | 36.7 | 1.0 | 59.3 | **98.7** | 64.7 | 9.3 | 37.0 | 45.3 | 5.3 | 17.3 | 2.3 | 41.7 | 61.7 | 6.0 | 34.7 |
| Lavie | 55.0 | 70.0 | 6.0 | 32.7 | 28.7 | 5.3 | 44.3 | 80.0 | **99.0** | 4.7 | 31.0 | 59.7 | 3.3 | 6.0 | 2.3 | 49.0 | 72.3 | 2.7 | 36.2 |
| ModelScope | 10.0 | 7.0 | 0.0 | 40.0 | 8.0 | 81.7 | 92.7 | 81.7 | 28.0 | **100** | 28.3 | 6.3 | 7.3 | 12.0 | 91.7 | 42.3 | 12.0 | 41.3 | 38.4 |
| NOVA | 51.3 | 68.3 | 5.7 | 71.0 | 54.0 | 2.3 | 70.3 | 71.7 | 55.3 | 4.7 | **99.0** | 61.0 | 0.3 | 3.3 | 8.7 | 42.3 | 34.3 | 12.7 | 39.8 |
| Pyramid | 96.0 | 93.0 | 11.7 | 27.7 | 47.3 | 0.7 | 19.7 | 73.3 | 89.3 | 0.7 | 66.0 | 0.3 | **99.3** | 3.0 | 15.0 | 3.3 | 72.0 | 5.7 | 45.1 |
| T2V-Zero | 30.3 | 13.0 | 0.0 | 3.7 | 1.7 | 26.3 | 76.0 | 84.3 | 77.7 | 26.0 | 1.0 | **98.7** | 64.3 | 77.0 | 4.3 | 52.3 | 38.7 | 0.7 | 37.6 |
| Tune-A-Video | 5.7 | 7.0 | 0.0 | 3.0 | 2.0 | 4.3 | 56.7 | 64.3 | 20.3 | 25.3 | 0.7 | 32.0 | 18.0 | **98.0** | 7.7 | 13.3 | 29.0 | 6.3 | 21.9 |
| VideoCrafer1 | 20.7 | 30.0 | 0.0 | 55.0 | 23.0 | 66.0 | 65.3 | 62.7 | 56.7 | 97.7 | 53.0 | 9.7 | 1.7 | 9.0 | **99.3** | 68.0 | 25.3 | 61.0 | 44.7 |
| VideoCrafer2 | 59.0 | 80.0 | 1.0 | 75.7 | 39.3 | 23.0 | 72.0 | 93.7 | 96.0 | 26.3 | 66.3 | 69.7 | 9.7 | 25.7 | 50.0 | **99.3** | 77.3 | 31.0 | 55.3 |
| Wan2.1 | 66.0 | 82.0 | 11.3 | 14.0 | 24.3 | 0.3 | 17.3 | 67.7 | 78.3 | 1.3 | 19.3 | 64.0 | 0.7 | 8.3 | 0.3 | 40.3 | **98.3** | 4.3 | 33.2 |
| ZeroScope | 39.3 | 48.7 | 1.3 | 48.7 | 33.7 | 0.3 | 25.0 | 36.0 | 44.7 | 34.0 | 36.7 | 24.7 | 8.7 | 23.3 | 52.7 | 43.7 | 23 | **99.3** | 34.7 |
| ChatGLM | 50.7 | 59.0 | 4.3 | 16.0 | 24.0 | 1.0 | 13.7 | 51.0 | 65.7 | 0.3 | 33.3 | 53.0 | 0.0 | 0.7 | 3.0 | 20.0 | 59.0 | 2.3 | 25.4 |
| Gen3 | 58.0 | 60.0 | 0.8 | 12.2 | 9.6 | 0.4 | 19.5 | 52.6 | 50.0 | 1.0 | 12.9 | 67.9 | 4.0 | 12.9 | 1.6 | 31.5 | 64.5 | 1.8 | 25.6 |
| Genmo | 75.7 | 82.7 | 1.3 | 39.0 | 31.0 | 1.0 | 20.7 | 70.3 | 75.3 | 2.0 | 55.7 | 81.3 | 4 | 8.3 | 9.0 | 67.0 | 75.3 | 23.7 | 40.2 |
| Hunyuan | 46.7 | 55.3 | 1.7 | 11.3 | 13.7 | 0.0 | 11.3 | 37.3 | 44.0 | 1.0 | 11.0 | 50.0 | 1.0 | 3.3 | 0.0 | 35.3 | 67.3 | 2.7 | 21.8 |
| Hailuo | 48.3 | 47.3 | 2.3 | 9.0 | 7.7 | 0.3 | 17.0 | 27.7 | 41.7 | 0.3 | 7.0 | 55.7 | 3.3 | 7.7 | 1.0 | 28.3 | 64.0 | 0.7 | 20.5 |
| Jimeng | 51.3 | 34.3 | 1.0 | 10.7 | 4.3 | 0.0 | 3.0 | 15.3 | 19.0 | 0.0 | 4.0 | 46.7 | 1.0 | 4.0 | 1.7 | 16.3 | 41.7 | 2.3 | 14.2 |
| Kling | 38.3 | 45.0 | 2.3 | 23.3 | 19.0 | 0.7 | 15.0 | 33.3 | 40.7 | 2.0 | 18.0 | 49.0 | 1.0 | 5.3 | 2.7 | 31.0 | 52.0 | 24.7 | 22.4 |
| Pixverse | 65.7 | 57.3 | 1.0 | 23.3 | 20.7 | 0.7 | 14.0 | 29.0 | 50.3 | 2.3 | 20.0 | 75.3 | 1.0 | 7.7 | 3.3 | 52.3 | 56.0 | 9.3 | 27.2 |
| Sora | 47.3 | 40.0 | 1.3 | 14.3 | 11.7 | 0.7 | 12.3 | 29.0 | 35.3 | 1.3 | 10.7 | 43.7 | 1.7 | 9.0 | 2.7 | 20.7 | 67.7 | 6.0 | 19.7 |
| Vidu1.5 | 60.7 | 54.7 | 2.0 | 4.7 | 10.3 | 2.0 | 14.3 | 38.0 | 45.3 | 1.3 | 11.3 | 63.7 | 1.7 | 11.3 | 4.0 | 44.0 | 57.7 | 1.7 | 23.8 |
| Wanxiang | 60.7 | 72.7 | 4.0 | 19.7 | 19.7 | 0.3 | 25.3 | 53.0 | 54.0 | 1.0 | 26.3 | 65.3 | 1.0 | 7.7 | 0.7 | 43.7 | 81.3 | 1.7 | 29.9 |
| Xunfei | 58.7 | 41.3 | 1.7 | 22.0 | 15.0 | 0.0 | 6.0 | 23.3 | 23.3 | 0.0 | 26.3 | 61.0 | 0.0 | 2.3 | 1.3 | 31.3 | 28.7 | 5.0 | 19.3 |
| Avg Acc | 47.0 | 52.1 | 6.4 | 28.3 | 24.8 | 11.4 | 37.6 | **57.3** | 53.3 | 16.8 | 30.0 | 50.5 | 7.4 | 15.1 | 16.1 | 39.7 | 51.8 | 13.9 | 31.1 |

- LoRA Rank (`lora_rank`): 16
- LoRA Alpha (`lora_alpha`): 16
- **Optimizer Settings:**
  - Weight Decay (`weight_decay`): 0.01
  - Warmup Ratio (`warmup_ratio`): 0.03.

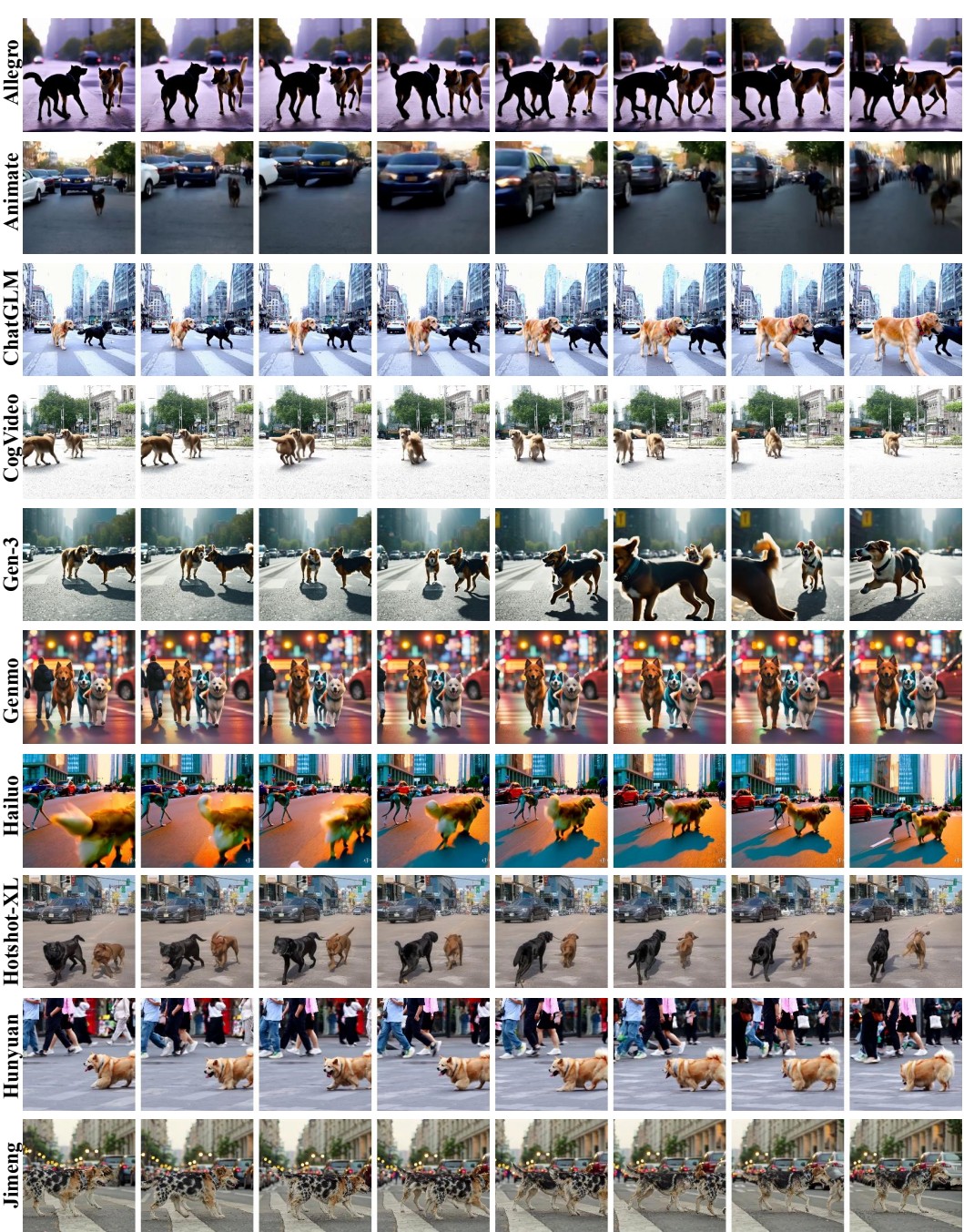

Figure 5: Visualization of the fake video frames in the FVBench dataset generated by different text-to-video generation models with the prompt "two dogs walk across a busy street".

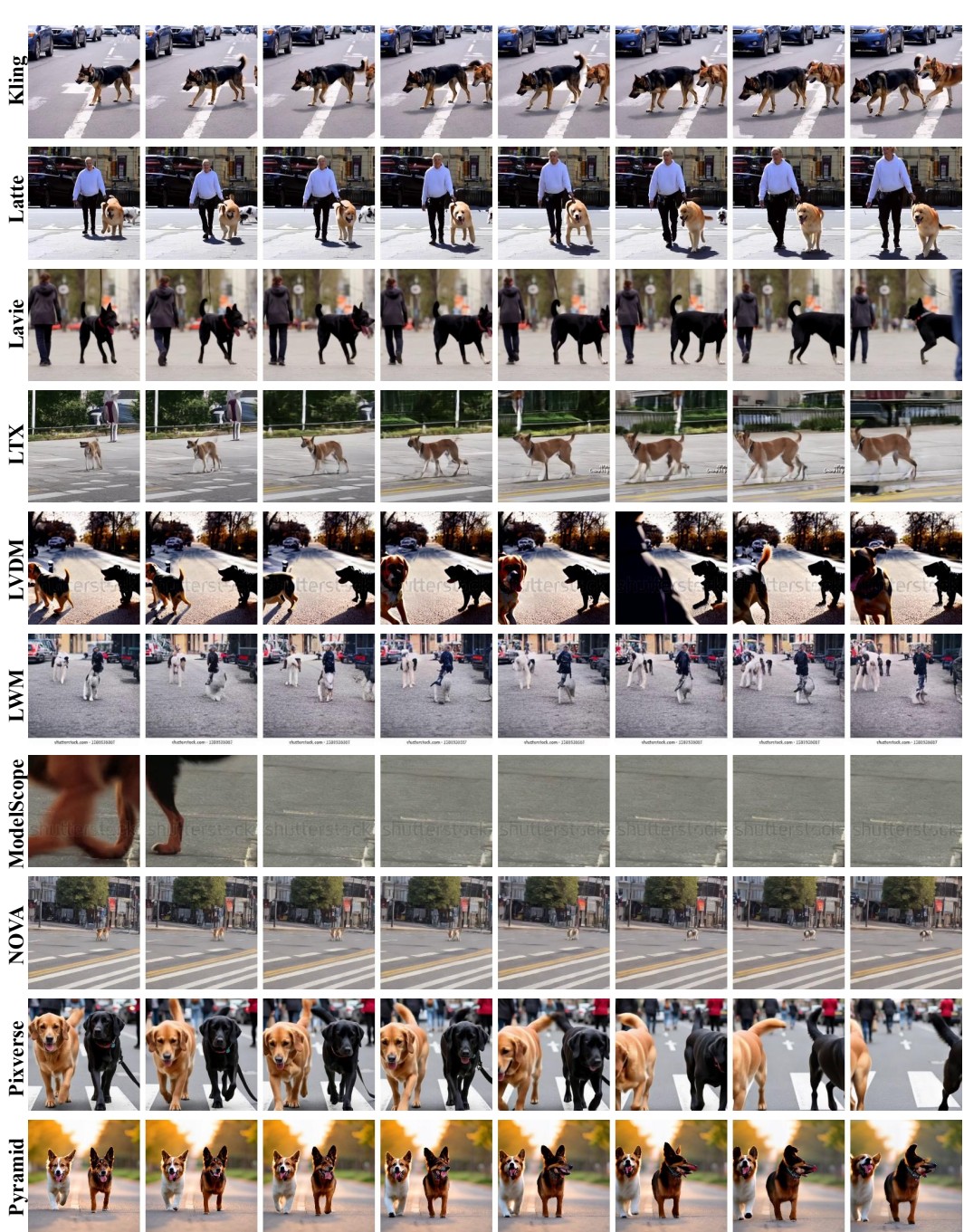

Figure 6: Visualization of the fake video frames in the FVBench dataset generated by different text-to-video generation models with the prompt "two dogs walk across a busy street".

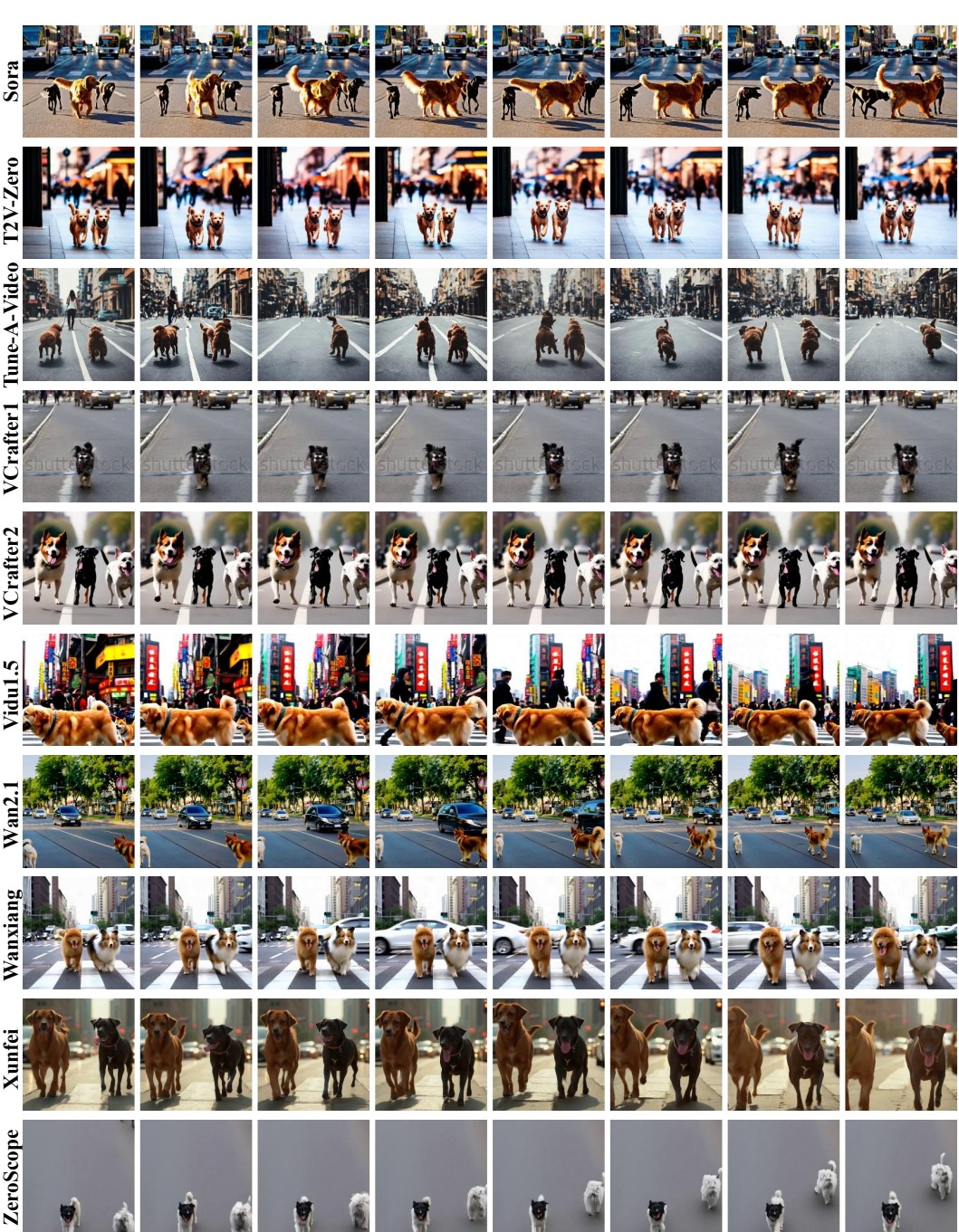

Figure 7: Visualization of the fake video frames in the FVBench dataset generated by different text-to-video generation models with the prompt "two dogs walk across a busy street".

