# OpenReview forum: "FVBench：Benchmarking Deepfake Video Detection Capability of Large Multimodal Models"
_ICLR.cc/2026/Conference — ICLR 2026 Conference Withdrawn Submission_

### Official Review · Reviewer_3jun · 2025-10-23

**Soundness:** 3
**Presentation:** 2
**Contribution:** 2
**Rating:** 4
**Confidence:** 5

**Summary:**

The deepfake forensics community constantly requires updated and large-scale datasets to advance the development of effective forensic methods. With this in mind, this paper introduces a new benchmark featuring approximately 120k videos created by around 40 AI-generated models. Additionally, the paper evaluates the performance of recent large language models (LLMs) on this dataset, both in-domain and across different domains, trying to investigate their potential in deepfake forensics. Specifically, it includes real videos from eight widely recognized public natural video datasets. For the AI-edited fake category, the authors collect 180 base videos from two existing datasets, which are then modified using 12 diffusion-based tools with prompts generated through DeepSeek. For AI-generated videos, 30 models are used, with prompts refined using DeepSeek R1. While this benchmark offers a valuable contribution to the community, the current version has several shortcomings that need to be addressed.

**Strengths:**

This paper makes a contribution to the deepfake forensics community by introducing a large-scale benchmark comprising approximately 120k videos generated by 40 AI models. It evaluates the performance of recent large language models (LLMs) in both in-domain and cross-domain scenarios. To ensure diversity, this paper uses real videos from eight datasets, and constructs both AI-edited fakes, created by modifying 180 base videos with 12 diffusion-based tools, and AI-generated videos, created by 30 models with prompts refined using DeepSeek. This dataset could advance deepfake detection methods, and give a insight for the zero-shot detection research.

**Weaknesses:**

First of all, the paper does not provide a comprehensive review of existing deepfake video datasets, overlooking several important works such as WildDeepfake (MM’20), Celeb-DF(++) (CVPR’20), OpenForensics (ICCV’21), and AV-Deepfake1M (MM’24). These datasets are critical for understanding the evolution of data diversity and authenticity in deepfake research.

Secondly, regarding the AI-edit category, I am concerned that relying solely on diffusion-based generative models may limit the dataset’s diversity. Although diffusion models have recently gained popularity, GAN-based and other generative approaches remain widely used across online media and social platforms. Incorporating these models would therefore help maintain representativeness and enhance the diversity of forgery sources in the benchmark.

Thirdly, the paper compares LLM-based and non-LLM-based deepfake detectors but does not provide proper citations or references for the compared methods. This omission makes it difficult to validate the results and assess the fairness of the comparison. Based on my knowledge and experience, methods such as Swin3D-T and ResNet3D-18 were introduced several years ago and are no longer considered state-of-the-art. Thus, their inclusion in the in-domain and cross-domain evaluations (Tables 5 and 6) may not convincingly demonstrate the advantages of LLM-based approaches. From the reported results, the performance gains of LLMs over Swin3D-T are relatively marginal (around 3%), which further limits the claimed impact.

To more effectively highlight the potential of LLMs for deepfake detection, I suggest incorporating more advanced and recently proposed detectors, such as UCF (ICCV’23), IID (CVPR’23), LSDA (CVPR’24), FreqBlender (NeurIPS’24), ProDet (NeurIPS’24), ForAda (CVPR’25), and Effort (ICML’25). Evaluating against these recent and competitive baselines would provide a stronger and fairer comparison.

Lastly, to make the benchmark more comprehensive and impactful, it would be valuable to go beyond evaluation of existing methods. The contribution could be significantly strengthened if a novel LLM-based detection framework were proposed and shown to outperform recent deepfake detectors in both in-domain and cross-domain scenarios. Such an addition would better demonstrate the superiority and practical value of the proposed benchmark.

**Questions:**

NA

---

### Official Review · Reviewer_wZsd · 2025-10-31

**Soundness:** 3
**Presentation:** 3
**Contribution:** 2
**Rating:** 4
**Confidence:** 5

**Summary:**

The paper introduces FVBench, a large-scale benchmark to evaluate deepfake/AI-generated video detection, with a particular emphasis on testing large multimodal models (LMMs) on this task. It aggregates ~120K videos from 8 real-video sources, 30 video generation models, and 12 video editing models, and reports a very wide comparison across classical detectors and many recent LMMs. The main empirical finding is that zero-shot generalization to unseen generators is still the bottleneck: both classical detectors and LMMs can hit high performance once fine-tuned, but performance drops notably in zero-shot and cross-generator settings.

**Strengths:**

1. The paper is easy to follow.
2. Timeliness and coverage. The authors explicitly include recent, powerful, and closed-source video generators (Kling, Sora, Gen-3, Vidu 1.5, Pixverse, Hailuo) that are very often missing in older deepfake datasets; that makes the benchmark much more reflective of what people will see in 2025+.
3. Public release promise. Dataset + code to be released upon publication helps reproducibility.
4. Clear empirical conclusion. The paper convincingly shows that “if you let me finetune, I can get to 100%, so the real problem is zero-shot and cross-generator generalization” — this is a good, clear takeaway for future work.

**Weaknesses:**

1. Contribution leans on scale and engineering work. Most ingredients (real-video sources, video editors, text-to-video models, prompt-based LMM evaluation) already exist (especially, the similar dataset curation procedure is also presented in previous work like [1,2,3]); the paper’s main step is to aggregate them and run a very broad eval. That is valuable but somewhat incremental for ICLR.
2. Prompt leakage / memorization. Since many videos are generated from common prompt tools, could detectors cheat with prompt-style artifacts instead of true visual artifacts? Any experiment with caption-free evaluation?
3. More discussion of recent related work like [1,2,4] could help strengthen the submission.
4. No deeper LMM formulation or analysis. The paper shows that prompting LMMs works to some extent and fine-tuning works well but it stops there. There is no proposal of other design. This part seems to be isolated from the motivation and could be an engineering exploration without enough novelty or say, deep analysis which could benefit the community.

Reference:
1. Distinguish Any Fake Videos: Unleashing the Power of Large-scale Data and Motion Features. 2024
2. How Far are AI-generated Videos from Simulating the 3D Visual World: A Learned 3D Evaluation Approach. 2025
3. DeMamba: AI-Generated Video Detection on Million-Scale GenVideo Benchmark. 2024
4. BusterX: MLLM-Powered AI-Generated Video Forgery Detection and Explanation. 2025

**Questions:**

My main concerns are about the academic contribution of this submission. I appreciate the efforts from the authors to build such a huge dataset and engineering framework including models and evaluation. However, in this submission, the academic contribution seems not sufficiently clear. All my questions are listed in the weaknesses part and I would like to increase my ratings if the authors' response could convince me.

---

### Official Review · Reviewer_TRF8 · 2025-10-31

**Soundness:** 3
**Presentation:** 3
**Contribution:** 3
**Rating:** 6
**Confidence:** 5

**Summary:**

This paper proposes FVBench, a large-scale benchmark for deepfake video detection with 120K videos covering real, AI-edited, and fully AI-generated categories from 42 SOTA video synthesis and editing models. The authors also propose a benchmark for large multimodal models, which finds that the zero-shot generalization ability to unseen generation models is the major challenge for current large models.

**Strengths:**

1. This paper proposes a very large-scale benchmark, including real videos from multiple sources, and AI-generated/edited fake videos from 42 SOTA generation models. The multiple sources, different categories, and mainstream generation models benefit the related community.
2. This paper evaluates the detection performance of existing large multimodal models, including almost mainstream models, which should also benefit the related area and give insight to other researchers.
3. The authors present a statistical analysis of real, AI-generated, and AI-edited videos in Fig.3, which is sound and interesting.
4. The experiments are actually extensive.

**Weaknesses:**

1. The authors state that they selected two of the LMM to fine-tune with LoRA. Can the authors present more details about fine-tuning? This should be helpful for understanding the results.
2. The authors evaluate the zero-shot performance in Tab.4 and Tab.5. Did they consider the few-shot ability of LMMs? Since providing few-shot samples should improve the performance, and this is low-cost compared to fine-tuning by LoRA.
3. From the results in Tab.5 and Tab.6, we can see the models exhibit strong intra-distribution performance but very poor generalization. Compared to the fine-tuned models in Tab.3 and Tab.4, where the fine-tuned InternVL can achieve nearly 100% accuracy and F1 score. Can I assume the fine-tuned models are overfitting? More justifications are helpful for analyzing this. If it is acually overfitting, did the authors consider how to solve this issue since there will always be generative models proposed in the future?
4. Did the authors consider evaluating some fake image/video detection methods? That will make this work more comprehensive.
5. I am curious about how long/how much the authors spent/cost on the generation and model evaluation. And did the authors double-check the quality of selected real or generated videos? This is important for constructing a benchmark. For the AI-edited videos, do the authors provide corresponding masks for the edited parts?

**Questions:**

I do appreciate the time and workload for constructing this large-scale video benchmark. I believe this work could be beneficial for the community but I still have some concerns about the experiments and datasets. So I currently lean towards a borderline accept ratings, I will change my ratings upon the authors' rebuttal.

---

### Official Review · Reviewer_yfLr · 2025-11-02

**Soundness:** 2
**Presentation:** 3
**Contribution:** 2
**Rating:** 2
**Confidence:** 4

**Summary:**

Existing deepfake detection techniques rely on datasets with limited generation methods, leading to concerns about their generalization performance for content generated by the latest generative models. The authors utilized 30 sota generative models and 12 editing models, constructing the FVBench dataset comprising 120,000 videos, including 8 real videos. They benchmarked the deepfake detection capabilities of LMM using the FVBench dataset.

**Strengths:**

1.	The study benchmarks Large Multimodal Models (LMMs) on a dataset created with a diverse set of the latest generative and editing models.

2.	It correctly recognizes the critical role of cross-generator evaluation in deepfake detection and provides a systematic assessment of generalization capabilities.

3.	The work presents a systematic evaluation of the deepfake detection capabilities inherent in LMMs.

**Weaknesses:**

1.	The paper points out the generalization limitations of existing deepfake detection models, and LMMs also appear to have poor zero-shot performance. Furthermore, existing models may suffer from pretrained data leakage shouldn't they be trained and tested under equivalent conditions?

2.	Is this demonstrating that zero-shot performance is crucial and can be resolved through fine-tuning? What is the rationale for using the LMM model? Additionally, include the state-of-the-art AIGC detection models (e.g., Demamba...) that have undergone fine-tuning in Tables 2 and 3.

3.	It is a predictable fact that performance can be improved by fine-tuning models on diverse datasets, and this has been extensively reported in existing papers. The conclusion asserted by the authors that using diverse datasets is necessary to enhance generalization performance is an intuitive fact.

4.	The authors numerically demonstrated the performance of LMMs but lacked a section for detailed analysis. They failed to provide a causal explanation for why the observed results in the experiments yielded such performance.

5.	If LMMs were used, I believe an analysis of explanatory power through text generation is necessary.

6.	It is necessary to analyze the characteristics of the generative models used. Since generative models often share similar techniques, I think this analysis is additionally required.

**Questions:**

Additional experiments and explanations are needed regarding the points mentioned above the weaknesses.

---

### Note · Authors · 2025-11-14

I have read and agree with the venue's withdrawal policy on behalf of myself and my co-authors.